# Uncovering actionable trade-offs of antifungal resistance in a yeast pathogen

Juan Carlos Nunez-Rodriguez [ID] [1,2], Miquel Àngel Schikora-Tamarit [ID] [1,2] & Toni Gabaldón [ID] [1,2,3,4] ✉

## Abstract

The increasing prevalence of antifungal resistance represents a major clinical challenge. To explore potential new therapeutic avenues, we investigated fitness trade-offs associated with azole and echinocandin resistance in *Nakaseomyces glabratus* (syn. *Candida glabrata*), a priority yeast pathogen showing growing incidence of drug and multidrug resistance. For this, we comprehensively phenotyped a large collection ($n = 77$) of azole- and echinocandin-resistant strains to uncover resistance-associated stress sensitivity trade-offs. Our results show that increased stress sensitivity is a common trade-off of drug resistance in this species, with 98% of resistant strains exhibiting reduced fitness under at least one of six assayed stresses. Despite the diversity of genetic backgrounds and resistance mechanisms represented by our collection, we identified consistent trends in some resistance-associated vulnerabilities. Using multivariate modeling we uncovered complex genetic interactions underlying these trade-offs. As a proof of concept for therapeutic potential, we experimentally validated the inhibitory effects of targeting some fitness trade-offs. Cyclosporin A selectively inhibited anidulafungin-resistant strains, while NaCl effectively suppressed the emergence of fluconazole resistance. This study highlights the widespread occurrence of fitness costs associated with antifungal resistance and emphasizes their potential as a novel therapeutic strategy against this growing threat.

**Keywords** Antimicrobial Resistance; *Nakaseomyces glabratus*; Fitness Trade-offs; Large-Scale Phenotyping; Novel Therapeutic Strategies
**Subject Category** Microbiology, Virology & Host Pathogen Interaction

## Introduction

A century after their introduction, the efficacy of antimicrobials is being undermined by a dramatic increase in antimicrobial resistance (AMR), considered among the most important global health threats facing humanity (Dadgostar, 2019). This scenario brings the urgent need not only to find new anti-infective compounds but also to understand the evolutionary process of drug adaptation so that strategies to overcome or prevent resistance can be adopted (Gabaldón, 2023). Given the limited portfolio of antimycotic compounds and the increased incidence of AMR, fungal pathogens such as those causing candidiasis are of particular relevance (Lass-Flörl et al, 2024). Annually, *Candida* species cause around 1.5 million bloodstream infections, leading to 995,000 deaths (Denning, 2024). While *Candida albicans* remains the most common source of candidiasis, non-albicans species such as *Candida auris*, *Candida parapsilosis*, and *Nakaseomyces glabratus* (formerly *Candida glabrata*) have become increasingly prevalent, currently accounting for nearly 50% of infections. Notably, *N. glabratus* is the second most common cause of candidiasis in many regions, and exhibits a high frequency of azole and echinocandin resistance (Lass-Flörl et al, 2024), posing a significant therapeutic challenge.

While the emergence of drug resistance is an expected evolutionary outcome of sustained selective pressure exerted by antimicrobial treatments, evolutionary theory indicates that adaptation to one trait is likely to compromise others (Garland, 2014; Stearns, 1989). This phenomenon, known as fitness trade-off, is a result from conflicts in resource allocation, molecular functions, or biochemical pathways, and typically manifests as reduced reproductive fitness under certain conditions. Fitness trade-offs have been demonstrated in diverse organisms and extensively studied in microorganisms (Zera and Harshman, 2001; Ferenci, 2016; Eames and Kortemme, 2012). For instance, experiments in *Escherichia coli* have shown that adaptation to high temperatures incurred a fitness cost at lower temperatures (Bennett and Lenski, 2007). Understanding fitness trade-offs can also pave the way to innovative therapies, particularly by exploiting collateral sensitivities (CS), where resistance to one drug increases susceptibility to another (Chowdhury et al, 2025). Examples include *E. coli* mutants resistant to tigecycline via *lon* mutations, which display increased sensitivity to nitrofurantoin (Roemhild et al, 2020), or *trkH*-mediated aminoglycoside resistance, which generates CS to chloramphenicol, β-lactams, tetracycline, and doxycycline (Lázár et al, 2013). This principle has inspired antimicrobial combination strategies aimed at preventing resistance and improving treatment efficacy (Baym et al, 2016; Munck et al, 2014; Lázár et al, 2022; Imamovic and Sommer, 2013). Conversely, the exploration of resistance trade-offs in the context of pathogenic fungi remains limited, with only a few studies underscoring the significance of trade-offs in the adaptation to antifungal drugs. In *Cryptococcus*, a clear correlation between resistance and fitness trade-offs has been described for mutations in *FCY1*, which drive resistance to 5-fluorocytosine (Després et al, 2022). In *C. albicans*,

[1]Barcelona Supercomputing Centre (BSC-CNS), Plaça Eusebi Güell, 1-3, 08034 Barcelona, Spain. [2]Institute for Research in Biomedicine (IRB Barcelona), The Barcelona Institute of Science and Technology, Baldiri Reixac, 10, 08028 Barcelona, Spain. [3]Catalan Institution for Research and Advanced Studies (ICREA), Barcelona, Spain. [4]Centro de Investigación Biomédica En Red de Enfermedades Infecciosas (CIBERINFEC), Barcelona, Spain. ✉E-mail: toni.gabaldon@bsc.es

antifungal combinations were shown to reduce resistance acquisition while imposing fitness trade-offs on resistant strains (Hill et al, 2015). Similarly, trade-offs have been suggested to constrain *C. albicans* evolution to amphotericin B resistance (Vincent et al, 2013). More recently, a comprehensive CS map in *C. auris* demonstrated that amphotericin B-resistant strains exhibit collateral sensitivity to multiple echinocandins, enabling the development of drug-cycling regimens that successfully prevented the emergence of resistance in vitro (Carolus et al, 2024a). However, despite these promising findings, our knowledge about resistance trade-offs in most pathogenic fungi remains poor, and the limited availability of antifungals restricts the applicability of combination therapies exploiting CS trade-offs. Hence, in the context of an escalating challenge of antifungal resistance, there is an urgent need to understand the impact and bases of drug resistance trade-offs, assessing their diversity as well as their potential applicability in drug discovery.

Here, we conducted a comprehensive analysis of possible trade-offs of drug adaptation by phenotyping a well-characterized collection of drug-adapted *N. glabratus* strains and their drug-sensitive parentals (Ksiezopolska et al, 2021). This collection was generated in a previous study by in vitro evolution in the presence of different drugs using 12 parental strains, representing the broad diversity of *N. glabratus* (Carreté et al, 2018). Genomic analyses revealed that these strains acquired resistance and multidrug resistance through clinically relevant mechanisms, such as mutations in transcription factors like *PDR1* and in drug targets like *ERG11* or *FKS1/2*. Such analyses also uncovered chromosomal duplications and novel mechanisms, such as ERG3-mediated cross-resistance. Importantly, several of the newly identified mutations and mechanisms have been subsequently identified in clinical studies (Scott et al, 2023; Carolus et al, 2025). Crucially, this collection provided us with the unique opportunity to directly compare phenotypes between drug-adapted strains and their naive parentals, enabling us to pinpoint phenotypic changes specifically caused by the acquisition of resistance, an analysis that is not feasible using unrelated clinical isolates. To chart fitness trade-offs of the different drug resistance mechanisms in this large collection, we used the Q-PHAST spot-assay methodology (Nunez-Rodriguez et al, 2025) to quantify growth under a battery of stressors targeting different cellular processes. Our results showed that the emergence of fitness trade-offs is a common feature of drug adaptation in *N. glabratus*, but also indicated that this outcome is highly heterogeneous, often comprising strain- and mechanism-specific traits. Nevertheless, we identified some common trends and the underlying genetic drivers. Finally, we provide a first proof of concept showing that trade-offs can be effectively actioned to limit the emergence of resistance in *N. glabratus*. Altogether, this study sheds light on the mechanisms underlying drug resistance-associated fitness trade-offs, outlines methods for their identification, and paves the way to develop innovative strategies to leverage these trade-offs for combating the growing challenge posed by AMR.

## Results

### Drug resistance leads to stress sensitivity trade-offs

To chart potential fitness trade-offs associated with the acquisition of drug resistance, we used large-scale quantitative phenotyping on a panel of stress conditions to characterize a previously generated collection of *N. glabratus* clinical strains and their drug-resistant descendants (Ksiezopolska et al, 2021) (Fig. 1).

This collection, comprises drug susceptible clinical strains from genetically diverse clades and their descendants obtained after directed in vitro evolution in anidulafungin (ANI), fluconazole (FLZ), a combination of both antifungals (ANIFLZ) and a drug-free control condition (YPD). Subsequently, strains evolved on ANI were evolved in fluconazole (AinF) and strains evolved on FLZ were evolved on anidulafungin (FinA) (Fig. 1A). These strains have been previously characterized in terms of their drug resistance profiles and the mutations acquired during the evolution experiments.

Here we profiled this collection in terms of their stress phenotypes. Aiming to cover alterations in processes typically affected by antifungal drugs, including cell wall, membrane and redox homeostasis, we selected six stress conditions (Fig. 1B): $H_2O_2$ (oxidative stress), DTT (reducing stress), CFW and CR (cell wall stress), SDS (membrane stress), and NaCl (osmotic/ionic stress). We used Q-PHAST (Quantitative Phenotyping and Antimicrobial Susceptibility Testing) (Nunez-Rodriguez et al, 2025), which is an accurate large-scale spot-assay system that uses image-analysis to generate growth curves and diverse fitness measurements (Fig. 1C).

To detect overall fitness trade-offs associated with the acquisition of resistance, we compared the fitness of each drug-resistant strain in rich medium using the "Area Under the Curve" (AUC) of each evolved strain divided by the AUC of its drug susceptible parental strain, referred to as the "AUC Relative" ($AUC_R$). In addition, to assess specific fitness defects under stress conditions, we calculated the "fitness AUC" (fAUC), which is the ratio of AUC under stress versus rich media conditions, to finally calculate the " fAUC Relative" ($fAUC_R$), which is the ratio of fAUC of the evolved strain and its corresponding parental strain (Fig. 1D). We used changes in $fAUC_R$ to identify fitness trade-offs in stress sensitivity, as this value controls for differences across strain backgrounds and fitness defects observable in rich media (i.e., not specifically associated with the stress condition).

We observed a generally lower fitness in YPD of the drug-evolved strains relative to their parental strains, with all evolved strains in all conditions being statistically different to the parental strains (Wilcoxon $p < 0.05$, see Methods), with the exception of YPD-evolved strains (Wilcoxon $p = 0.764$) (Fig. 2C and Dataset EV1). Strains adapted to the two drugs simultaneously (ANIFLZ) showed the largest growth defects ($AUC_R$ median 0.665, MAD 0.074) (Fig. 2C,D), while YPD-evolved strains exhibited similar or slightly higher fitness than their parental strains ($AUC_R$ median 0.906, MAD 0.043). These findings indicate that a decrease in fitness in rich medium is common in drug-resistant strains, although not universal.

Notably, a reduction in fitness was apparent for all drug-resistant strain groups in all stress conditions, except for DTT, where we found an opposite trend (Fig. 2A). To validate this unexpected finding, we repeated the screen in liquid medium with three different concentrations of DTT (5, 6, 8 mM) and confirmed, albeit with some variation, that many strains were more resistant to DTT than their parental strains (Dataset EV2). We hypothesized that drug resistance mechanisms could result in higher levels of intracellular reactive oxygen species (ROS) thereby rendering the presence of the reducing agent beneficial. To test this, we measured intracellular ROS levels in selected wild type and resistant strains to

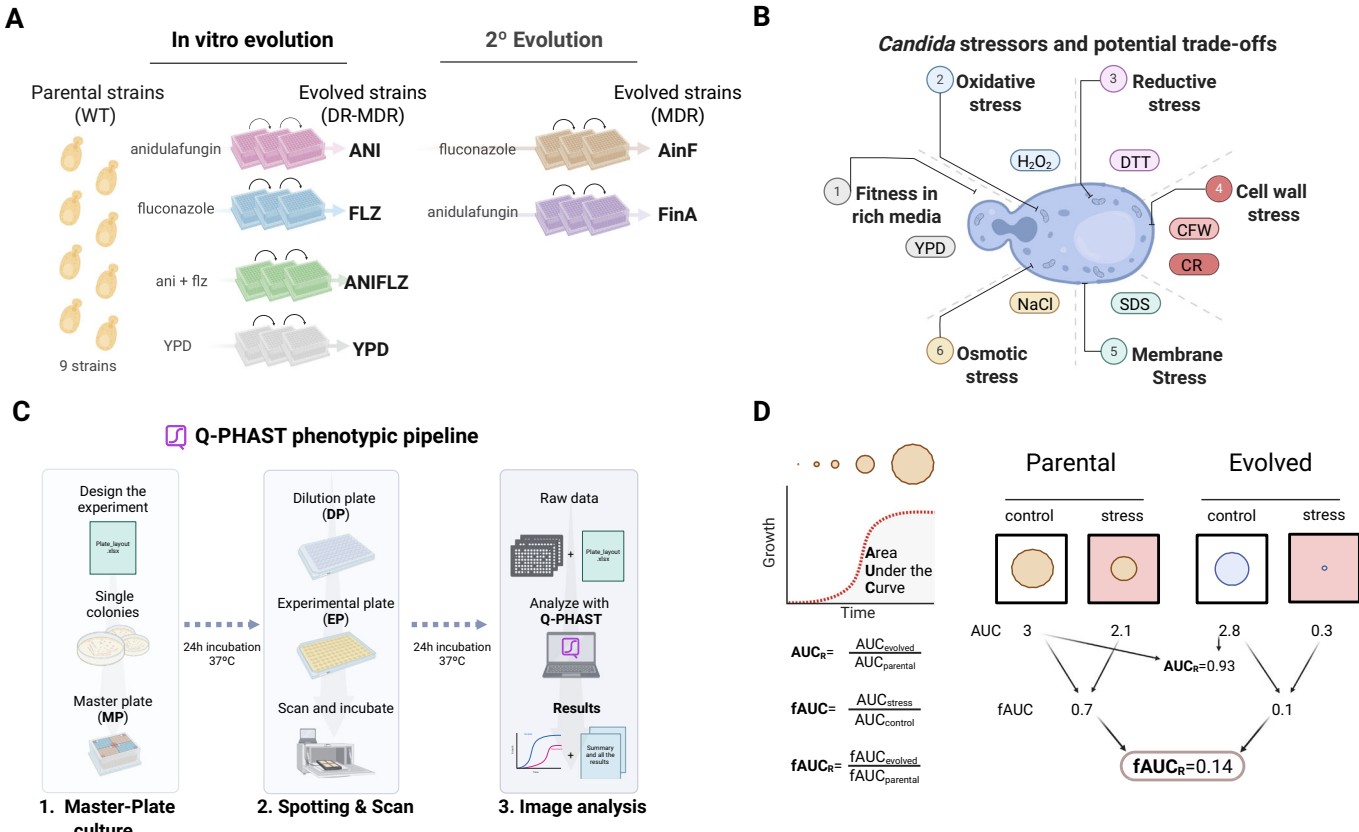

**Figure 1. Integrated approach for detecting trade-offs.**

(**A**) Graphical summary depicting the origin of the drug-resistant strain collection from Ksiezopolska et al (2021) used in this study. In the first round of in vitro evolution, the ANI, FLZ, ANIFLZ, and YPD strain groups were obtained. In the second round of evolution, starting from the ANI strains, AinF strains were derived, and from the FLZ strains, FinA strains were derived. (**B**) Stress conditions used for the detection of trade-offs. (**C**) Overview of the Q-PHAST methodology pipeline. On day 1, the experiment is designed, and the Master-Plate (MP) is cultured. On day 2, the Dilution-Plate (DP) is prepared to spot the strains onto the Experimental-Plate (EP), which is then incubated on scanners that capture images of strain growth. On day 3, the images are analyzed using Q-PHAST software to generate results. (**D**) Calculations for detecting trade-offs. Growth curves are used to compute the area under the curve (AUC) for each strain, which is then employed to calculate the AUCR, fAUC and fAUCR. The graphical representation illustrates the calculations performed to derive these factors. YPD medium was used as control and stressors are indicated in (**B**) and in Methods.

determine if higher intracellular ROS was associated with resistance. However, although some resistant strains showed higher levels of intracellular ROS than their parentals, we did not observe a clear pattern (Dataset EV3).

For all stress conditions, we found some drug-resistant strains showing no growth defect compared to their parental strains (near the red dashed line, Fig. 2A). However, most of the strains did exhibit growth defects (median $fAUC_R < 0.9$, one-sided t-test (compared to 1) $p < 0.05$), including severe ones (median $fAUC_R < 0.5$, one-sided t-test (compared to 1) $p < 0.05$) such as complete growth inhibition (Fig. 2A,B, Dataset EV4). In all drug-evolved groups, we found statistical differences from their parents in response to at least four different stressors, whereas this did not occur in YPD-evolved groups (wilcoxon_p_diff_from_1 column in Dataset EV1). The largest negative trade-offs were found under NaCl, YPD and CFW (Figs. 2A,B and EV1).

Interestingly, we identified stress trade-offs that tend to correlate with specific resistance profiles (Fig. 2A). For instance, anidulafungin-resistant strains (ANI, AinF, FinA, ANIFLZ) tend to show the highest trade-offs under cell wall (CFW, CR) and

membrane (SDS) stress conditions. This was particularly apparent for FinA strains under CFW stress. While fluconazole-resistant strains (FLZ) showed mild defects under these conditions, their anidulafungin-evolved progeny (FinA) clearly showed stronger defects, similar to those of ANI strains. In contrast, fluconazole-resistant strains were more specifically affected by NaCl. With FLZ strains showing higher costs than ANI strains, and ANI strains subsequently adapted to fluconazole (AinF) decreased fitness under this condition.

Notably, 98% of the resistant strains exhibited a significant trade-off in at least one of the tested conditions (Fig. 2E), with most strains having between two and four trade-offs (Fig. 2F). The most frequent combinations of these trade-offs involved fitness defects in NaCl+YPD, followed by the combination of trade-offs in NaCl+YPD + CFW + CR (Fig. EV1). Although this phenotypic profiling encompassed the full diversity of *N. glabratus* and various resistance-conferring mutations to azoles and/or echinocandins, we sought to determine if stress-related trade-offs could be detected in a drug-resistant strain isolated in a clinical setting. A general limitation of clinical strains is the lack of a naive parental strain,

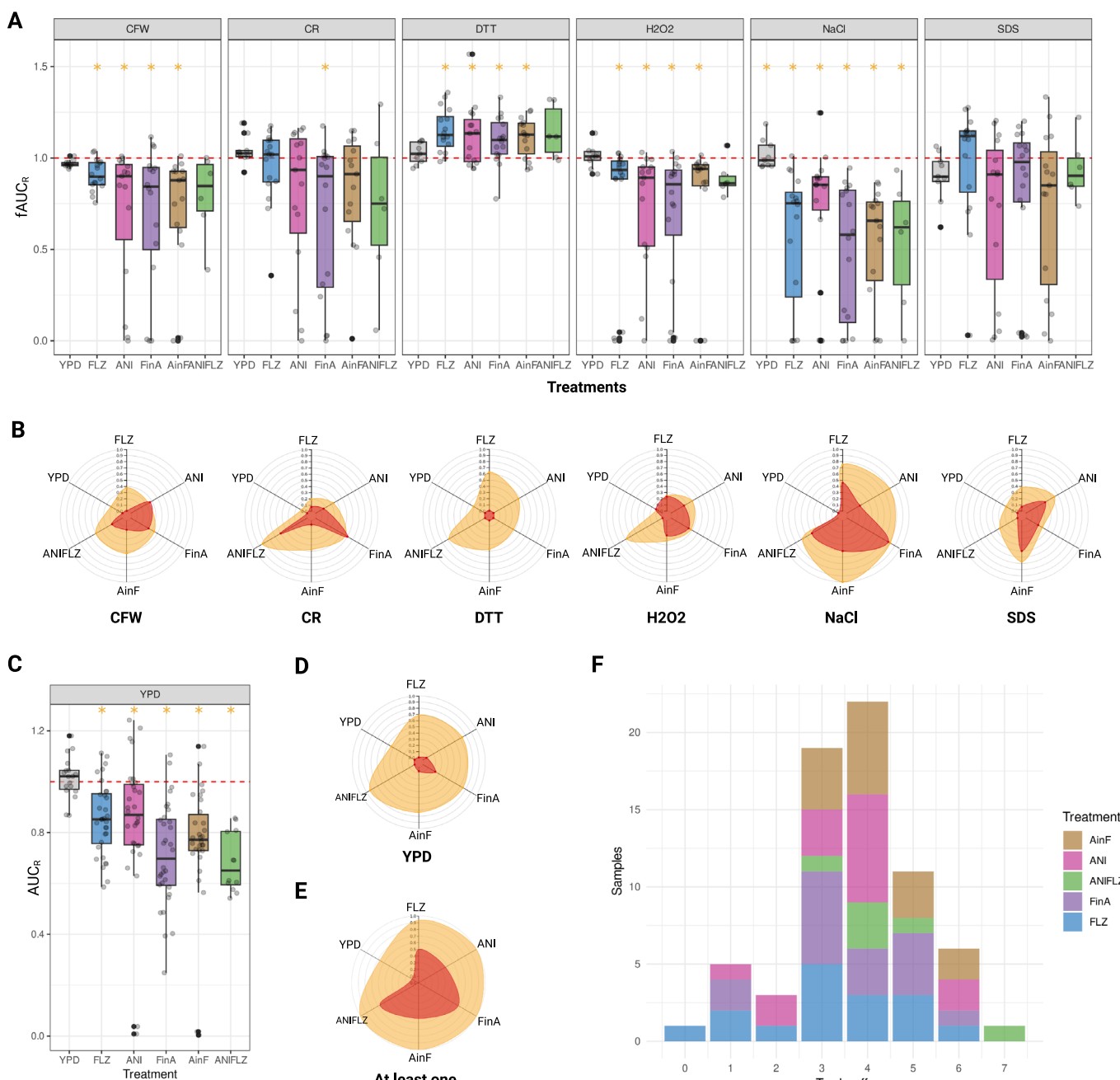

**Figure 2. Stress response trade-offs associated with antifungal resistance.**

(A, C) Box plots showing the distribution of fAUC$_R$ (stress fitness relative to their wild type) or AUC$_R$ (rich media fitness relative to their wild type) for each strain evolved under different treatments in the different stress conditions. For each box plot, the central line represents the median, the box bounds correspond to the first (Q1) and third quartiles (Q3), and the whiskers extend to the most extreme data points within 1.5× the interquartile range (IQR) from the quartiles. Dots outside the whiskers represent outliers. The red dashed line marks the wild-type reference (fitness = 1). Each dot within the box plot indicates the median fitness value of four biological replicates for a given strain. Orange asterisks indicate statistically significant differences ($p < 0.05$) between the evolved strains and the wild type, as determined by a two-sided Wilcoxon test (see Dataset EV1 for data and exact $p$-values for each condition). (B, D) Radar plots showing the frequency of strains grouped by treatments that are statistically different from their wild type (orange) and those that show severe trade-offs (red). (E) Radar plot showing the frequency of having at least one stress trade-off of each strain grouped by treatments. (F) Histogram showing the number of statistically significant trade-offs for each strain, with colors indicating the evolution treatment. Source data are available online for this figure.

which would preclude accurately assessing fitness effects of the resistance-conferring mutations, given the absence of a comparable naive reference and the extremely large variability observed among wild-type strains. To circumvent this problem we exploited the availability of a well-characterized pair of serial clinical isolates, DSY562 and DSY565, where the latter is an azole-resistant derivative obtained from the same patient after 50 days of fluconazole therapy (Vale-Silva et al, 2017). Phenotypic comparisons using YPD and three of the stresses (CFW, $H_2O_2$, and NaCl) revealed that fitness in YPD was similar between the two clinical strains, with the resistant strain having a slightly lower $AUC_R$ value of 0.876, MAD 0.068 (Dataset EV4, Treatment "Clinical-FLZ"). Similarly, no significant differences were observed under CFW ($fAUC_R$ = 0.951, MAD = 0.030) or $H_2O_2$ ($fAUC_R$ = 1.066, MAD = 0.020). However, in line with previous findings in our collection, we observed a significant loss of fitness (median $fAUC_R < 0.9$, one-sided t-test (compared to 1) $p < 0.05$) under NaCl stress ($fAUC_R$ = 0.634, MAD = 0.115). This supports the broader trend of NaCl sensitivity as a consistent trade-off associated with fluconazole resistance in *N. glabratus*. Altogether, these results provide compelling evidence that defects in the stress response are a prevalent trade-off in *N. glabratus* drug-resistant strains.

## Targeting trade-offs to combat resistance acquisition

The concomitant appearance of resistance and associated trade-offs opens potential new avenues to combat the emergence of drug resistance. In particular, by targeting the trade-off during drug exposure, the appearance of resistance might be reduced. To provide a proof of concept to this principle, we focused on NaCl stress and fluconazole treatment, as the most significant and commonly observed trade-off in our experiment.

We first performed a competition assay between wild-type and multidrug-resistant strain in either YPD or YPD supplemented with 1.5 M NaCl and counted colony-forming units (CFU) in plates containing YPD (where both strains grow) or YPD+anidulafungin (where only the multidrug-resistant strain grows). Our results (Fig. EV2) showed that the wild-type strain outcompeted the multidrug-resistant strain in both media, with the multidrug strain being totally absent when NaCl was present. Additionally, we performed a competition assay in vivo using the *Galleria mellonella* larvae infection model, where the competitive disadvantage of the multidrug-resistant strain was also observed (Fig. EV2). Hence, in the absence of drug pressure, the wild type strain out-competes the multidrug-resistant strain, due to the fitness trade-offs present in the latter. Importantly, this disadvantage is enhanced in the presence of NaCl, which exacerbates fitness trade-offs, leading to faster clearance of the multidrug-resistant strain.

We next performed a checkerboard assay, which revealed a synergistic interaction between fluconazole and NaCl. While the wild type strain grew similarly to the control at fluconazole concentrations up to 16 µg/ml and at NaCl concentrations of 1.1 M, the combination of 16 µg/ml fluconazole with a lower NaCl concentration (0.21 M) almost completely inhibited growth (Fig. EV3A,B). This finding, together with the observation that NaCl is a primary trade-off of fluconazole resistance, suggests that the mechanisms required for an appropriate response to NaCl overlap with those involved in the antifungal activity of fluconazole in *N. glabratus*. Consequently, to

achieve fluconazole resistance, these mechanisms may be affected, resulting in defects in the NaCl response.

We next reasoned that, if fluconazole resistant strains frequently have defects in the NaCl response, then the presence of this stressor during fluconazole exposure may reduce the probability of the acquisition of resistance. The rationale behind this approach was that any selected strain would need to overcome two bottlenecks: acquiring drug resistance while avoiding the typical trade-off of altered responsiveness to NaCl (Fig. 3A). To test this, we plated $10^7$ cells on YPD agar, YPD supplemented with 1.25 M NaCl, YPD with 128 µg/mL fluconazole, and YPD containing both NaCl and fluconazole. After 72 h, no colony-forming units (CFUs) were detected under the combined treatment in any of the three biological replicates, whereas single-stressor plates showed confluent growth (Fig. EV3C). Consistently, when approximately 500 fluconazole-adapted colonies were replica-plated in NaCl, only two were able to grow on NaCl, while all YPD-adapted colonies grew on the same concentration of NaCl (Fig. EV3D). These results support the previously identified fitness trade-off in our resistant isolate collection and suggest that applying a double bottleneck based on trade-off constraints may genetically limit the selection of resistant strains.

Finally, to investigate whether this trade-off could be exploited to limit the emergence of resistance over time, we developed a standardized Adaptability assay (see Methods). This novel test consists of a 4-day directed evolution experiment that assesses the ability to adapt to certain conditions, which we here used to assess ability to adapt to fluconazole in the presence or absence of NaCl (Fig. 3B). In brief, replicate populations are exposed to a gradient of drug concentrations and each day the best adapted population (MGC) is used as seed for the following day where the same conditions are tested. As the experiment progresses, adapting populations are expected to grow at increasingly higher drug concentrations, with the highest concentration to which strains can adapt over four days serving as a proxy for their adaptive potential. This can be compared with and without the presence of NaCl to calculate whether this compound accelerates or delays adaptation.

In the absence of NaCl, CBS138 was able to grow at the maximum drug concentration (256 µg/ml) in 4 days, with an $ADP_{(4)0.5-256}$ value of 32 (indicating a 32-fold increase under the experimental conditions). In contrast, in the presence of 1.25 M NaCl, the results were markedly different. The strains failed to adapt to fluconazole and maintained their initial susceptibility ($ADV_{(4)0.5-256}$ = 1) despite 4 days of fluconazole exposure (Fig. 3C,D). These results were consistent across four biological replicates and also when using the alternative wild-type strain BG2, that belongs to another clade (Fig. 3C, Dataset EV5). This clearly shows that NaCl severely reduces adaptation to fluconazole as hypothesized.

Altogether, these results highlight the potential of trade-offs as actionable targets, not only to eliminate resistant strains but also to reduce the likelihood of resistance development.

## Cyclosporine A selectively inhibits anidulafungin-resistant strains by targeting its trade-offs

The emergence of trade-offs in response to different stressors suggests that shared underlying mechanisms may be affected. Signal transduction pathways are key for sensing and responding to

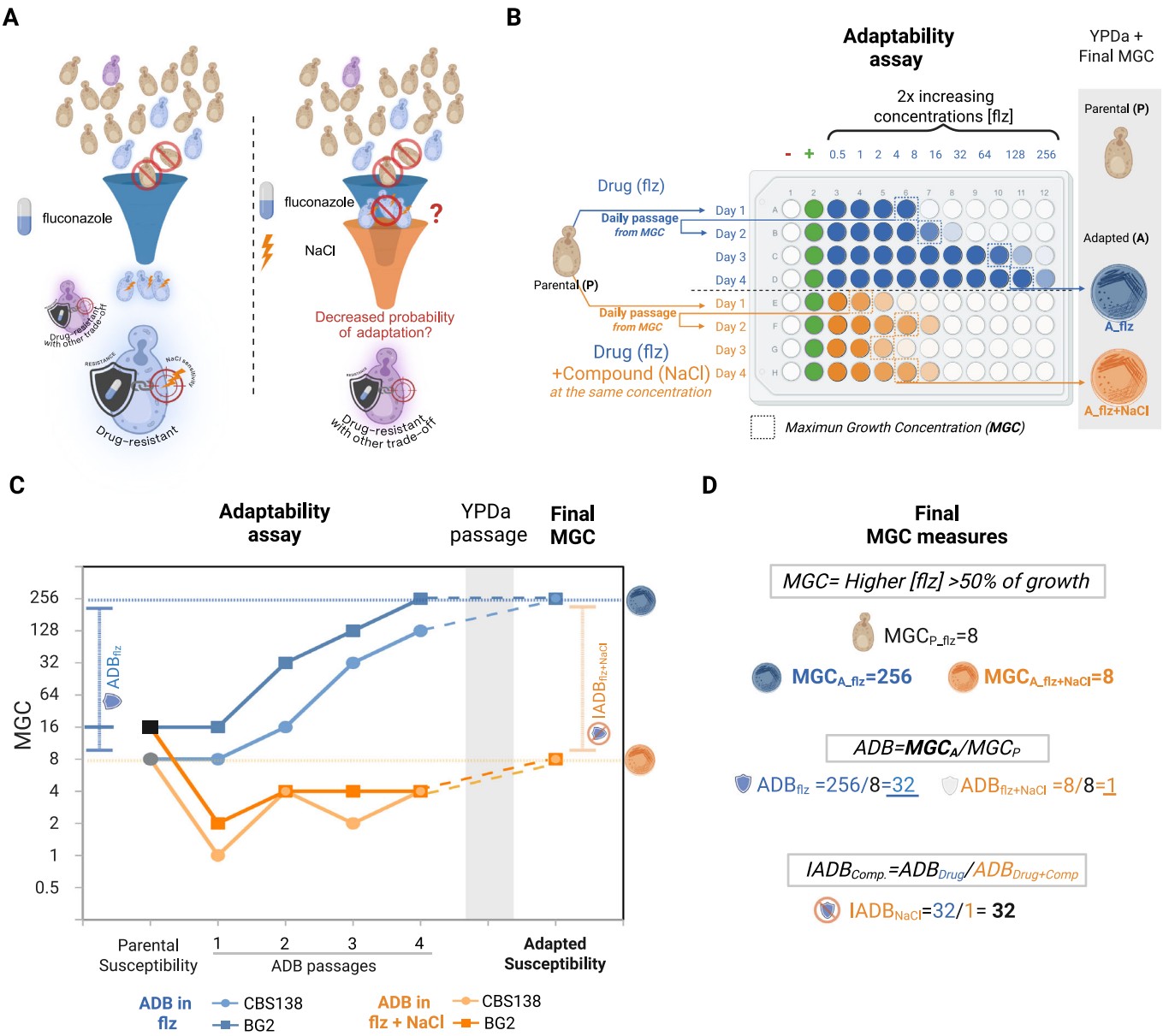

**Figure 3. NaCl reduces adaptability to fluconazole.**

(A) Conceptual scheme for the hypothesis that NaCl acts as a double bottleneck to reduce adaptability to fluconazole (flz). (B) Experimental design scheme to evaluate adaptability (ADB) under selective pressure from fluconazole. Populations are exposed daily to 10 different fluconazole concentrations, from the MGC population (Maximum Growth Concentration, defined as the maximum drug concentration at which growth exceeds 50% of the control population). Parallel experiments are conducted with and without NaCl to assess its effect on ADB. After 4 series of daily passages, an intermediate incubation in YPD medium for 72 h is performed, followed by susceptibility testing of adapted (A) strains to be compared with its parental (P) strains. (C) Line plot showing fluconazole susceptibility, measured as MGC, CBS138 and BG2 wilt type reference strains with and without NaCl during the experiment. (D), Final MGC measurements, along with calculations of ADB and inhibition of adaptability (IADB) values. Representative populations are displayed in (B), (C) and (D); full datasets, including all replicates and measurements, are presented in Dataset EV5. Source data are available online for this figure.

multiple environmental stresses, with the calcineurin signaling pathway being central to multiple stress responses (Miyazaki et al, 2010; Yadav and Heitman, 2023). This pathway can be inhibited by the commercially available drug cyclosporin A (CsA), which induces hypersensitivity to cell wall stress and alters the antifungal response (Sanglard et al, 2003). Previous studies have shown that some strains with mutations affecting these responses are dependent on this pathway (Pavesic et al, 2024). We hypothesized

that the use of CsA may allow the selective elimination of resistant strains based on their stress-related trade-offs. To test this, we performed CsA dose–response studies using the CBS138 wild-type strain and their ANI, FLZ, AinF and FinA evolved strains (Fig. EV4). The results showed that the wild type, FLZ and FinA strains could grow at high concentrations of CsA (up to 60 μM). In contrast, ANI and AinF strains showed high sensitivity to CsA (MIC50 around 1.9 μg/ml).

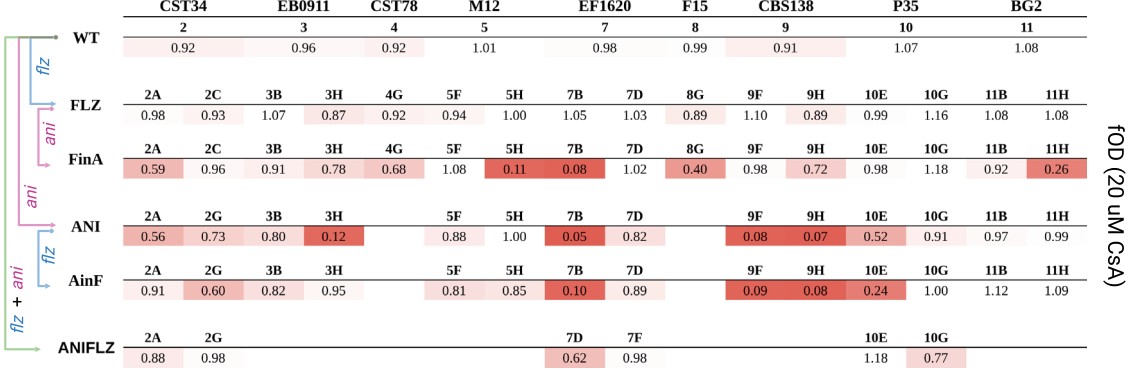

**Figure 4. Cyclosporine A activity on drug-resistant strains.**

Susceptibility table showing the relative fitness (fOD) in Cyclosporine A (CsA) 20 μM compared to control medium of the collection of drug-resistant strains. Growth is shown on a color scale where white represents 1 (no-affected by CsA) and red represents 0 (highly affected by CsA). Source data are available online for this figure.

To assess whether CsA sensitivity was strain-specific or it was shared among resistant strains, we screened all resistant strains, measuring their growth in liquid YPD and YPD supplemented with 20 μM CsA. The results indicated that 45% ($n = 31$) of the resistant strains showed increased sensitivity to CsA (median $fOD_R < 0.9$, one-sided t-test (compared to 1) $p < 0.05$, see Dataset EV4 for comparisons), with 19% ($n = 13$) showing a severe trade-off (median $fOD_R < 0.5$, one-sided t-test (compared to 1) $p < 0.05$). In contrast, all wild-type strains showed normal growth under these conditions (Fig. 4). Importantly, increased CsA susceptibility was predominantly observed in anidulafungin-resistant strains (13/15 ANI strains with a median increase of 33% of susceptibility and 5 of them with severe trade-offs). Whereas only two of sixteen FLZ strains showed changes in CsA susceptibility (with only a median susceptibility increase of 12%), this number increased to seven in their anidulafungin-evolved progenies (FinA, with a median susceptibility increase of 60%), four of which with severe increase susceptibility. Conversely, some ANI CsA-susceptible strains lost their high CsA sensitivity when evolved on fluconazole, as seen in the AinF 3H strain (Fig. 4). This may result from the loss of selective pressure for anidulafungin resistance, leading to the loss of mutations conferring both anidulafungin resistance and CsA susceptibility.

To demonstrate that CsA is able to eliminate multidrug-resistant strains in heterogeneous populations, we performed a competition assay in the presence of CsA, as done earlier for NaCl. The results from this experiment (Fig. EV2) indicated that the wild-type strain outcompeted the multidrug-resistant strain (AinF-9F) in the presence of CsA. In fact, no anidulafungin-resistant colonies were identified at the end of the experiment.

Taken together, these results demonstrate a critical role of the calcineurin pathway not only in stress and drug responses, but also in the survival of many anidulafungin-resistant strains, uncovering potential novel targets for such strains. It also proves the concept that it is possible to identify small molecules that specifically target trade-offs to selectively clear multidrug-resistant strains.

## Complex genetic interactions underlie fitness trade-offs of drug resistance

We subsequently used a statistical modeling approach to generate hypotheses about potential genetic mechanisms underlying the observed variability in trade-off intensity (TI, which we estimated from $fAUC_R$, $fOD_R$ or $AUC_R$, depending on the condition). In brief, we modeled variation in $AUC_R$ (for YPD), $fAUC_R$ (for stress conditions), or $fOD_R$ (for CsA) based on two types of strain features. First, we considered the sequence variants and aneuploidies acquired during in vitro evolution, which potentially underlie TI variation. Second, as different variants may have equivalent effects, we grouped them by either (i) affected gene (ii) region of affected genes or (iii) pathway of affected genes, and considered each 'group of variants' (e.g., missense mutations in some region of *FKS1*) as a different feature. Third, we considered several characteristics of the wild-type *N. glabratus* strain background: wild-type strain, AUC and/or fAUC of the wild type in a given condition. In total we considered 770 features (both binary and continuous) as potential predictors for TI (Fig. 5A).

We first used a univariate modeling approach, where we predicted TI from each of these features independently. We considered the proportion of variation ($r^2$) explained by a linear model based on the feature as a proxy for the strength of the genotype-phenotype association, and calculated its statistical significance (see Methods). With few exceptions, such models were either not significant ($p$ bonferroni-corrected $< 0.05$) or explained very little variation in TI ($r^2 < 0.1$) (Fig. 5B), suggesting that TI is mostly driven by a combination of genetic features (epistasis), which cannot be captured with such univariate modeling.

To better accommodate epistasis we built various multivariate models that considered all these features simultaneously. In brief, for each condition, we built a different model for (i) various statistical methods (multiple linear regression, random forest regression or regression trees), (ii) different feature selection models (e.g., forward selection or recursive feature elimination) and (iii) distinct parameters of the models (e.g., tree pruning parameters or the consideration of two-way interactions in linear models). In total, we built 58 models per condition, each of them yielding a minimal set of predictive features of TI variation, inferred with cross-validation (CV). To control for overfitting we calculated a model consistency score, which measures whether a model/feature selection approach yields similar features across alternative CV sets. We additionally inferred the empirical probability of observing equivalent models by chance, similar to

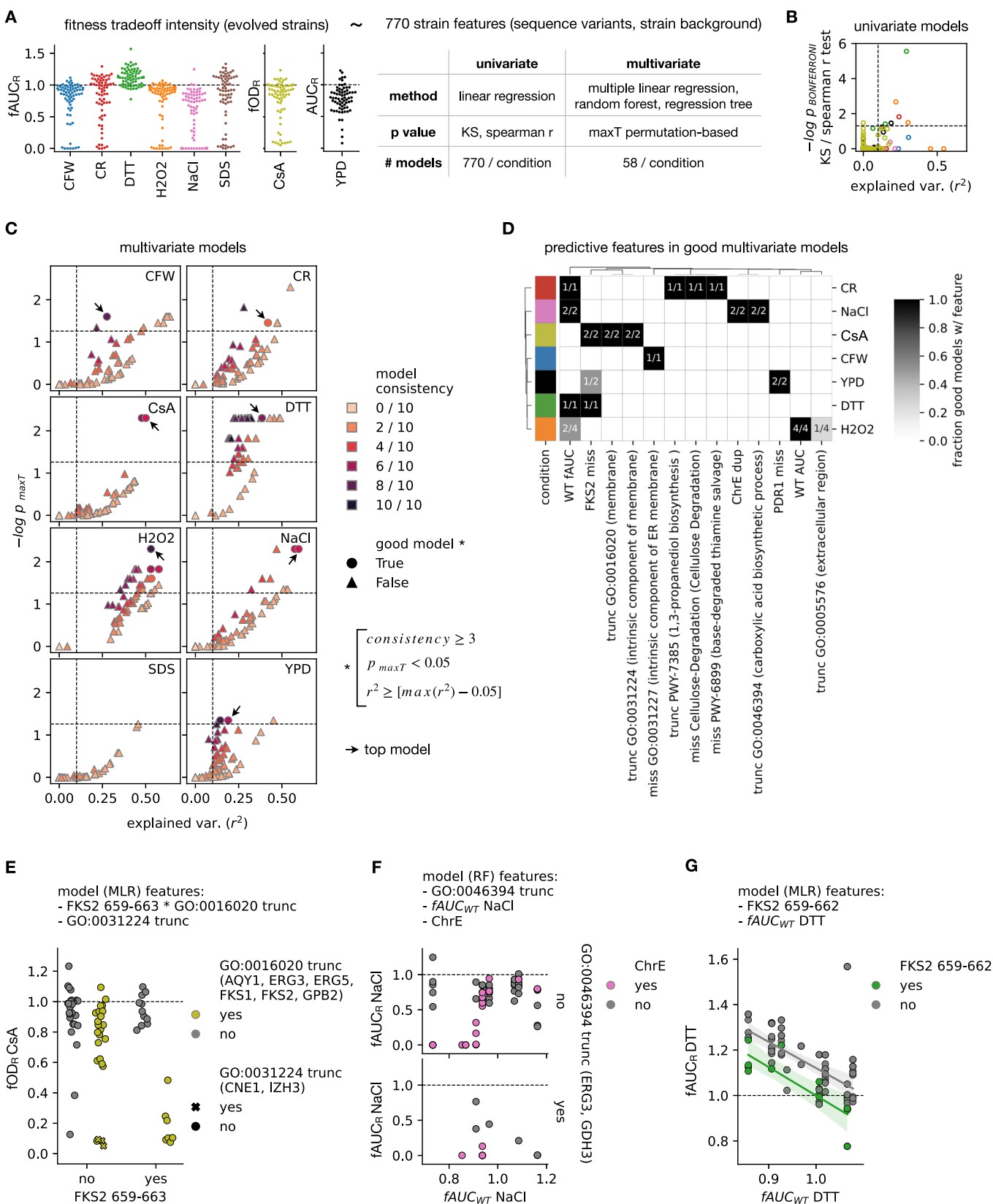

◀  **Figure 5.  Genome predictive models of trade-offs.**

(A) Pipeline followed to model the fitness trade-off intensity (TI) of each strain (left panels) from up to 720 features, related to the acquired variants during drug adaptation and intrinsic properties of the strain background (e.g., fitness in the stressor). To provide flexibility to our approach, we tried first performing a univariate analysis, modeling TI variation in a given condition from each individual feature. To calculate $p$ values for each feature we used either a KS test (binary features) or a Spearman $r$ test (continuous features). Also, we tried various multivariate modeling approaches, 58 in total, using multiple linear regression, random forests and regression trees. Also for each we calculated a maxT $p$ value reflecting the probability of observing a model with a given predictive performance ($r^2$) and model consistency by chance (see Methods). (B) For the univariate modeling, the explained variation ($r^2$) of each model vs the $-\log$ bonferroni-corrected $p$ value of the corresponding feature. The colors are equivalent to (A). (C) For each of the 58 multivariate models, the explained variation ($r^2$) vs the $-\log$ maxT $p$ value, where each panel is one condition. The color represents the model consistency, which reflects the number of random iterations (out of ten) of building the model that yielded the same predictive features. The symbols indicate which models were deemed as 'good' (maxT $p < 0.05$, consistency $> 2$ and $r^2$ above (max $r^2 - 0.05$). Also, the arrows indicate the top models used to explore the predictive features. (D) Heatmap showing, for each condition (rows), the fraction of good models that consider each of the predictive features (columns). (E–G) Scatterplots showing how different feature combinations, used in the top models, are related to TI values. Each panel corresponds to a condition (only conditions where we find acceptable predictive performance and relevant epistatic effects), and the list of features are those used by the model. For instance, the top model for CsA (E) uses a multiple linear regression (MLR) based on (i) the interaction between *FKS2* mutations in the protein region 659–663 and truncating variants in genes with "membrane" annotation (GO:0016020) and (ii) truncating mutations in genes with annotation "intrinsic component of membrane" (GO:0031224). Conversely, the top model for NaCl (F) uses a random forest (RF) based on (i) truncations in genes with annotation "carboxylic acid biosynthesis" (GO:0046394), (ii) fAUC in NaCl of the wild-type strain and (iii) chromosome E duplications. Finally, the top model for DTT (G) uses MLR based on (i) fAUC in DTT of the wild-type strain and (ii) *FKS2* mutations in the region 659–662; where we add a fitted line showing the trend for each group. Source data are available online for this figure.

the maxT GWAS approach (Gundlach et al, 2016; Schikora-Tamarit and Gabaldón, 2024), so that models with maxT $p < 0.05$ are unlikely to be overfit. Thus, for each condition we build 58 models, each with a particular set of features, $r^2$, consistency score and maxT $p$ value (Fig. 5C).

We could find relevant models (with $r^2 \geq 0.1$, $p$ maxT $< 0.05$, consistency $\geq 3$) for all conditions except SDS (Fig. 5C). To focus on the most important results we further investigated the features yielded by the best models for each condition (with $p$ maxT $< 0.05$, consistency $\geq 3$ and $r^2$ within the 95 percentile of all models). We found that such 'good models' considered (i) wild-type strain background features (fAUC and AUC), (ii) known drug resistance genes (*FKS2* and *PDR1* missense variants), (iii) chromosome E duplications and/or (iv) pathway alterations (membrane, ER-membrane, extracellular region, 1,3-propanediol biosynthesis, cellulose degradation, thiamine salvage and carboxylic acid biosynthesis) (Fig. 5D). While most of these features were condition-specific, some were relevant in multiple conditions (i.e., wild-type fAUC and *FKS2* missense variants), suggesting some shared fitness trade-off mechanisms. As these 'good models' mostly used similar features (Fig. 5D), we focused on a single 'top model' per condition. We defined this model as the 'good model' with highest $r^2$ and/or consistency score (Fig. 5C), which is particularly useful to understand how these features explain TI variation (Figs. 5E–G, EV5 and 6).

To further investigate trade-off drivers we visualized TI values split by the different predictive features of the top models (Figs. 5E–G and EV5). For NaCl, low TI values are correlated to (i) a combination of chromosome E duplications and low wild-type fitness in NaCl or (ii) truncations in *ERG3* or *GDH3*, related to carboxylic acid biosynthesis (Fig. 5F). Conversely, for DTT the wild-type fAUC in DTT is negatively correlated to TI (Fig. 5G), which may be explained because most strains had fAUC$_R > 1$, in contrast to the other stressors. Also, we find that *FKS2* hotspot missense variants generate lower DTT TI (Fig. 5G). For H$_2$O$_2$, the main driver of TI variation was the wild-type AUC in H$_2$O$_2$ (positive correlation, see Fig. EV5A), and we could not find clear mutational signatures explaining the remaining variation. For CFW, missense variants in ER membrane genes (*CNE1* and *KRE6*) were associated with low TI values (Fig. EV5B). For YPD low TI values are correlated to a combination of *FKS2* hotspot variants and

*PDR1* missense variants affecting amino acids in the region 280–768 (Fig. EV5C). Finally, for CR, the top model suggests that low TI values are correlated to either (i) a combination of missense mutations in genes related to thiamine salvage and genes involved in cellulose degradation, or (ii) a combination of truncations in genes related to 1,3-propanediol biosynthesis and lower wild-type fAUC (Fig. EV5D). All in all, these results suggest the genotype-phenotype relationships underlying TI variation.

For CsA our top model suggest that low TI are in strains with (i) a combination of *FKS2* missense variants in region 659–663 (overlaps hotspot 2) and truncations in membrane genes (including *ERG3, ERG5* and *FKS1*) or (ii) *CNE1* and *IZH3* truncations, with "integral component of membrane" annotations (Fig. 5E). However, this did not explain the low TI values for 3/13 strains with fOD$_R < 0.5$ (Fig. 5E), which motivated us to further explore the specific mutations found in these 13 strains (Dataset EV4). Notably, all of them had hotspot 2 missense variants in *FKS2* (between amino acids 657 and 708) and (mostly truncating) mutations in either (i) *FKS1* (5/13), (ii) *FKS1* and *ERG3* (5/13), (iii) *FKS1* and *CNE1* (1/13) or (iv) *CNE1* (2/13). This suggests that a combination of *FKS2* variants and further truncations in cell wall-related proteins (*ERG3, FKS1* and/or *CNE1* (Yutaka et al, 2018), see 'General Discussion' of Schikora-Tamarit (2023)) generate higher CsA susceptibility.

Beyond the specific results of each model, we can extract some general learnings from these analyses. First, *FKS2* hotspot mutations, which underlie ANI resistance, often generate trade-offs (in CsA, DTT and YPD), especially in combination with other mutations (i.e., in membrane proteins for CsA, or in *PDR1* for YPD). Second, combinations of acquired mutations rather than a single one are the main predictive features of TI, suggesting that complex epistatic interactions underlie various fitness trade-offs. Third, multiple linear regression including two-way interactions is a powerful way to capture these complex genotype-phenotype relationships, as this approach is used by most (6/7) top models (Figs. 5E–G and EV5). Fourth, fitness in the wild-type strains was often a relevant predictive feature, suggesting that interactions between acquired mutations and the strain genetic background are key determinants. Fifth, as seen for NaCl, the strain genetic background may determine the adaptive constraints imposed on chromosomal duplications, providing further insights into the role

of such complex variants in drug adaptation. In summary, these results deepen our understanding about the precise mutational processes leading to different fitness trade-offs of drug adaptation.

## Discussion

The acquisition of drug resistance is becoming increasingly prevalent in pathogenic yeasts such as *N. glabratus*. Although several recent studies have shed light onto the genetic bases of acquired resistance in this species (Lee et al, 2023; Berman and Krysan, 2020), we still have a very poor understanding of what other phenotypic effects, beyond resistance, these mutations may have. Previous studies have reported a long-term stability of genetically-acquired drug resistance (Ksiezopolska et al, 2024), and generally lower growth rates as compared to related susceptible strains (Ksiezopolska et al, 2021). However, these analyses used rich media and cannot capture specific trade-offs under stress conditions.

Drug resistance fitness trade-offs can be defined as the collateral consequences of drug adaptation that result in a decrease in fitness under certain conditions (Garland, 2014; Yekani et al, 2023). However, detecting these trade-offs from clinical, drug-resistant isolates is challenging, given the general unavailability of the drug-naive parental strain. In addition, resistance to the same drug may emerge from different genetic mechanisms or mutations, and the same mutation may have different impacts depending on the genetic background. To overcome these limitations we used a well-characterized collection of drug-resistant *N. glabratus* strains and their susceptible parentals, which provided us with a unique opportunity to directly compare fitness between resistant strains, and their susceptible parentals, whose genotypes differ in just a few known mutations. Our study focused on different stress conditions commonly associated with antifungal targets (cell wall and cell membrane) together with stresses relevant to infection (such as oxidative and reducing stress). To obtain such a comprehensive cartography of phenotypes across a large battery of strains we resorted to Q-PHAST (Nunez-Rodriguez et al, 2025), a recently-developed high-throughput spot phenotyping system that provided us with reproducible quantitative fitness data.

Our results indicate that drug-resistance is commonly associated with stress trade-offs in *N. glabratus*, with 98% of the drug-resistant strains having reduced fitness, at least in one stress condition, as compared to their susceptible parents. We observed that anidulafungin-resistant strains commonly show trade-offs in cell-wall, membrane and oxidative stresses, while fluconazole-resistant strains are more sensitive to osmotic stress. Unexpectedly, we also observed a higher relative increase in reductive stress in all resistant strains, suggesting an altered handling of reactive oxygen species in these strains that deserves further investigation. The occurrence of trade-offs during the acquisition of resistance was validated by phenotyping a pair of serially isolated clinical strains, in which it was observed that the strain that had developed resistance during treatment with fluconazole had defects in the osmotic stress response, validating that trade-offs also arise in clinical resistant strains. Recent experiments in *C. auris* have identified drug-associated fitness trade-offs, including fitness defects under osmotic stress associated to fluconazole resistance (Das et al, 2024); cell wall stress linked to echinocandin resistance (Jenull et al, 2022); and

multiple stresses linked to amphotericin B resistance (Carolus et al, 2024b). Similarly, higher sensitivity to stressors has been described in azole-resistant *C. parapsilosis* strains (Papp et al, 2020). Together with earlier studies reporting fitness loss in resistant *C. albicans* strains (Popp et al, 2017), these findings suggest that the emergence of fitness trade-offs associated with drug resistance may also be an expected evolutionary outcome in other *Candida* species. This idea should be confirmed with additional comparative studies including different *Candida* species.

We used different statistical analyses to associate phenotypic changes with acquired genetic variants. Our results show that the changes in phenotypes have complex underlying mechanisms involving epistatic relationships between different mutations and the genetic background of the strain. As a result, the same genetic alteration underlying the drug-resistant phenotype, can have diverse trade-offs in different genetic backgrounds, a complexity that might be exacerbated by the large genetic diversity encompassed by *N. glabratus* (Carreté et al, 2018). Despite this difficulty, we generated several multivariate models that could significantly predict the trade-off intensity and uncovered specific genes related to each phenotype. For instance, *FKS2* hotspot changes combined with other mutations often generate trade-offs. Also, a relevant finding is the observation that chromosome E duplications generate reduced fitness in NaCl depending on the original fitness of the strain. A limitation of our modeling approach is that it failed to capture the factors underlying a significant fraction of variation in trade-off intensity for CFW, YPD, SDS and CR. We speculate that this is likely due to limited sample size as well as to our limited understanding of the functional impacts of different variants (e.g., on protein stability or overall cell biology) that prevent us grouping them in more meaningful ways. Thus, there is still room for further research clarifying what underlies TI variation for these conditions. Similarly, larger sample sizes may help to clarify these trade-offs.

We hypothesized that stress sensitivity trade-offs could be exploited to combat drug-resistant strains and to prevent the emergence of resistance during antifungal treatment. As a proof of concept, we used NaCl, the stressor for which the most trade-offs were detected. Since more than 80% of the drug-resistant strains analyzed were more sensitive to NaCl than their parental counterparts, we reasoned that under simultaneous selective pressure from both the drug and the stressor, many of the resistance-conferring mutations would not have been selected due to their poor fitness. Hence, co-treatment could potentially reduce the number of resistant variants selected during drug exposure, due to counter-selection by the stressor. Our results provide evidence that the presence of NaCl during drug treatment reduces the emergence of resistance and that there is a clear synergistic effect between NaCl and fluconazole. Given the heterogeneous response among strains, the mechanistic connection underlying this effect appears complex, epistatic, and background-dependent. One possible contributing factor is that fluconazole exposure induces changes in membrane composition that may increase ion permeability, as described in *Candida albicans* (Kolecka et al, 2009)) Similar effects could result from resistance driving mutations in the ergosterol pathway (Eliaš et al, 2024). This changes in sterol composition, in turn, could reduce NaCl tolerance, as previously described in *S. cereviseae* (Kodedová and Sychrová, 2015), explaining the observed synergy between fluconazole and NaCl, and the reduced tolerance in fluconazole-resistant strains. Our models also point to a possible

link between chromosome E aneuploidy and NaCl sensitivity (Fig. 5F). These duplications, encompassing ERG11, aside from potentially affecting sterol composition, could additionally result in profound transcriptional remodeling (Mackey et al, 2025; Tsai and Nelliat, 2019; Tsai et al, 2019) that could negatively impact homeostasis. Moreover, drug- and stress-response signaling pathways are highly interconnected, including those mediated by calcineurin, protein kinase C, and the high-osmolarity glycerol (HOG) pathway (Iyer et al, 2022). Therefore, mutations conferring resistance may destabilize this delicate homeostasis balance in the membrane, making it more difficult to select mutations conferring drug resistance without simultaneously compromising stress responses. Although NaCl itself has no direct clinical relevance, our findings suggest that novel drugs mimicking its effects, or targeting key stress response mechanisms, could be exploited to achieve similar outcomes in antifungal therapy. In line with this idea, a recent unbiased combinatorial screening in *C. albicans* identified a compound targeting fungal membrane homeostasis (Revie et al, 2022) by inducing membrane-associated stress, proved to be highly effective in combination with fluconazole for the elimination of *Candida* both in vitro and in vivo.

With the aim of targeting trade-offs with drugs, we tested CsA, a well known and commercially available drug that acts by inhibiting the calcineurin pathway, which regulates several stress responses in *Candida spp.* (Sanglard et al, 2003). Hence, we hypothesized that, due to their stress-related trade-offs, drug-resistant strains would be calcineurin-dependent and therefore susceptible to this drug. Supporting this, we found that most anidulafungin-resistant strains were more susceptible to this CsA than their drug-sensitive parents (Fig. 4). To elucidate the genetic bases of this relationship, we examined the multivariate models relating genomic information with CsA trade-offs, particularly for variants found in anidulafungin-resistant strains exhibiting severe trade-offs (growth <50%), which suggested that a combination of *FKS2* hotspot 2 variants and additional truncations in cell wall-related proteins (*ERG3*, *FKS1* and/or *CNE1*) generated higher CsA susceptibility. When *FKS1* is compromised *FKS2* expression is necessarily increased to maintain cell wall biosynthesis (Katiyar et al, 2012), this is specifically regulated by the calcineurin pathway (Pavesic et al, 2024; Katiyar et al, 2012; Miyazaki et al, 2010). Resistance conferred by *FKS2* and its activity is highly expression-dependent (Garcia-Effron et al, 2009), which means that reducing it may impair its essential functions. Consistently, calcineurin inhibition partially reverses *FKS2*-mediated echinocandin resistance (Katiyar et al, 2012).

Given that catalytic activity of Fks2p and, more importantly, Fks1p is reduced by resistance-causing mutations (Yutaka et al, 2018; Garcia-Effron et al, 2009), we hypothesize that maintaining necessary levels of cell wall biosynthesis mutants strains in genes related with cell wall integrity requires increased *FKS2* activity, which is achieved through upregulation via the calcineurin pathway. Resistant strains with *FKS2* hotspot 2 mutations, combined with defects in other cell wall-related genes, likely become highly dependent on the calcineurin pathway to sustain the necessary level of cell wall biosynthesis. Inhibition of calcineurin by CsA disrupts this (now essential) pathway, leading to severe growth defects. Consistent with this hypothesis, recent studies in *N. glabratus* have identified cell wall biosynthesis as a process where disruption of genes such as *FKS1* induce calcineurin dependence (Pavesic et al, 2024). This has also been shown in *CNE1* mutants, which have altered cell wall structure with reduced β-1,6-glucan and increased accumulation of chitin (Pavesic et al, 2024; Yutaka et al, 2018). Such dependence on calcineurin results in demonstrated strong fitness defects in the presence of its inhibitors (Pavesic et al, 2024; Yutaka et al, 2018). Novel fungal-specific calcineurin inhibitors that avoid immunosuppressive effects (Hoy et al, 2022; Gobeil et al, 2021) could strategically exploit these actionable trade-offs against echinocandin-resistant strains. These findings reinforce the therapeutic potential of stress-related trade-offs as promising avenues for combating antifungal resistance.

Altogether, our study provides important insights into stress sensitivity trade-offs associated with different drug resistance mechanisms in *N. glabratus*. While such trade-offs are a ubiquitous consequence of drug adaptation, their manifestation is strikingly heterogeneous, with strain- and mechanism-specific characteristics. Importantly, we identified recurring patterns and the genetic underpinnings that drive these trade-offs. These trade-offs may explain why, in the absence of drugs, wild-type strains could be more competitive than multidrug-resistant strains, both in vitro and in vivo. In certain selective scenarios, this could lead to strains with unfavorable trade-offs being displaced by wild-type strains (Ben-Ami and Kontoyiannis, 2012; Ben-Ami et al, 2011; Imbert et al, 2016). To avoid this, resistant strains could favor variants that lose their resistance to increase their fitness or acquire complementary mutations to compensate for the fitness loss (Carolus et al, 2024b; Ksiezopolska et al, 2024; Eckartt et al, 2024). However, the emergence of additional mutations may lead to new fitness trade-offs. This suggests that although most strains exhibit trade-offs, they do not always impact multifactorial host competitiveness or infective capacity (Ferrari et al, 2009; Vale-Silva et al, 2013; Arastehfar et al, 2024). Thus, strains containing variants that are not detrimental in their selection environment are more likely to persist over time. Using the same reasoning, even if the fitness trade-offs are not important for the current selective pressure, they are present and therefore potentially targetable. This leads to the conclusion that strains that have just acquired resistance and strains that have undergone subsequent adaptations both have potential vulnerabilities. To this end, our study shows compelling evidence supporting that these trade-offs can be actionable targets in therapies directed to combat drug resistance and prevent the emergence of resistance. Ultimately, our work not only elucidates fitness trade-offs resulting from the acquisition of resistance, but also highlights their potential utility in the development of innovative strategies to combat the escalating challenge of antimicrobial resistance.

# Methods

**Reagents and tools table**

| Reagent/Resource | Reference or Source | Identifier or Catalog Number |
| --- | --- | --- |
| **Experimental models** | | |
| Nakasomyces glabratus drug resistant collection | Toni Gabaldón Lab (IRB Barcelona, Spain) https://doi.org/10.1016/j.cub.2021.09.084 | NA |

| Reagent/Resource | Reference or Source | Identifier or Catalog Number |
|---|---|---|
| Galleria mellonella | Ferry Hagen Lab (Westerdijk Fungal Biodiversity Institute, Utrecht, The Netherlands) https://doi.org/10.1111/eea.13237 | NA |
| **Chemicals, enzymes and other reagents** | | |
| Calcofluor White (CFW) | Sigma-Aldrich | F3543-1G |
| Congo Red (CR) | Sigma-Aldrich | C6277-25g |
| Dithiothreitol (DTT) | Sigma-Aldrich | 43817 |
| Hydrogen Peroxide (H2O2) | Sigma-Aldrich | H1009-100ML |
| Sodium Dodecyl Sulfate (SDS) | PANREAC | A2263,0100 |
| Sodium Chloride (NaCl) | PANREAC | A2942,1000 |
| Cyclosporine A (CsA) | MERCK | 30024-100 mg |
| 2′,7′-dichlorofluorescein diacetate (H2DCFDA) | MERCK | D6883 |
| **Software** | | |
| Q-PHAST | https://doi.org/10.1038/s41596-025-01179-z | v1 |
| Combenefit | https://doi.org/10.1093/bioinformatics/btw230 | 2.02 |
| Python | Welcome to Python.org https://www.python.org/. | 3.9.7 |
| scipy | Virtanen et al (2020) | v1.11.3 |
| numpy | Harris et al (2020) | 1.26.0 |
| biopython | Cock et al (2009) | 1.78 |
| pandas | McKinney (2010). https://doi.org/10.25080/Majora-92bf1922-00a. | 2.1.1 |
| statsmodels | Seabold and Perktold (2010). https://doi.org/10.25080/Majora-92bf1922-011. | 0.14.0 |
| scikit-learn | Pedregosa et al (2011) | 1.3.0 |
| matplotlib | Hunter (2007). https://ieeexplore.ieee.org/document/4160265. | 3.8.0 |
| seaborn | Waskom (2021) | 0.13.2 |
| Biorender | Biorender.com | Online version |
| **Other** | | |
| Platemaster | Gilson | PLATEMASTER 220 µL (F110762) |
| Scanners | EPSON | Epson Perfection V600 photo scanner (B11B198032) |
| Flow cytometer | Beckman Coulter | Gallios multi-color flow cytometer |
| Microplate Spectrophotometer | Thermo Scientific | Multiskan SkyHigh (A51119700DPC) |

## Fungal strains and culture conditions

We used a previously-generated and well-characterized collection of 77 drug resistant strains and their nine drug-sensitive parental clinical isolates (Ksiezopolska et al, 2021). In brief, these strains represent seven genetically diverse clades of *N. glabratus* (Carreté et al, 2018) and had been in vitro adapted to different conditions including a YPD control and YPD supplemented with increasing concentrations of anidulafungin (ANI), fluconazole (FLZ), the two drugs in combination (ANIFLZ), or in sequential treatments (AinF and FinA, Fig. 1A). We retrieved associated meta-data for these strains, including drug susceptibility, and genomic variants underlying potential mechanisms of resistance.

We grew the strains from $-80\,°C$, 20% glycerol stocks by streaking them on YPD-agar plates to obtain single-colonies, which were further grown in deep 96-well plates. Growth conditions were $37\,°C$ in YPD (1% yeast extract, 2% bactopeptone, 2% glucose) with or without agar (2%), to which the necessary study compounds were added before solidification.

## Large-scale quantitative stress phenotyping

We employed the Q-PHAST methodology (Nunez-Rodriguez et al, 2025) to phenotype the strain collection under various stress-inducing conditions including YPD as a control, and YPD supplemented with Calcofluor White (CFW, 25 µg/ml), Congo Red (CR, 125 µg/ml), Sodium Dodecyl Sulfate (SDS, 50 µg/ml), Hydrogen Peroxide ($H_2O_2$, 5 mM), Dithiothreitol (DTT, 7 mM), or Sodium Chloride (NaCl, 1 M).

We followed the standard Q-PHAST phenotyping protocol. In brief, 4 single colonies of each strain are grown for 24 h in YPD in a deep 96-well Master-Plate (MP) and serve as biological replicates. Once grown to culture saturation, a 3 µl dilution is made in 200 µl water to form the Dilution-Plate (DP), and 96 spots of 5 µl are placed on a solid medium plate containing the study substance, which we will refer to as Experimental-Plates (EPs). The EPs are incubated on the top of scanners in an incubator at $37\,°C$ and images are taken every 15 min for 24 h. These images, along with metadata, are analyzed using the graphical interface of the Q-PHAST software (available at https://github.com/Gabaldonlab/Q-PHAST/). This software analyzes the images to measure the growth of each spot over the 24 h period, resulting in a growth curve. The area under the curve (AUC) was calculated for each spot, which is used as a proxy for fitness in each of the conditions and from which the $AUC_R$ ($AUC_{evolved}/AUC_{parental}$), fAUC ($AUC_{stress}/AUC_{control}$) and $fAUC_R$ ($fAUC_{evolved}/AUC_{parental}$) are calculated (Fig. 1D). The results shown are the median from the 4 replicates.

## Screening assays in liquid medium

Screening in liquid medium was performed analogously to the phenotyping in solid medium described above. The Master-Plate (MP) was cultured for 24 h with shaking at $37\,°C$. Subsequently, 3 µL of the MP was transferred to the Dilution-Plate (DP), and 5 µL of this DP was transferred to a 96-well plate containing a homogeneous mixture of 200 µL of YPD and the relevant stress

components: DTT at 8 mM, 6.4 mM and 5.12 mM concentrations, or CsA at 20 µM. These plates were incubated at 37 °C for 24 h. After incubation, the wells were resuspended and growth was measured as a function of Optical Density at 600 nm (OD600). For each replicate, the "fitness OD" (fOD) in the test condition was calculated by dividing OD600 in the test condition by OD600 in YPD. Finally, the median and median absolute deviation were calculated for all strains and conditions. Also, to take into account the wild-type variation in fOD, we defined a $fOD_R$ measure, which is calculated as $fAUC_R$ (Fig. 1D) but using OD600 instead of AUC ($fOD_{evolved}/fOD_{parental}$).

## Dose–response assays

Fresh cells were resuspended in 1 ml of water and counted three times using a DeNovix CellDrop Automated Cell Counter, with the median count used to prepare a $2.5 \times 10^5$ cells/ml in YPD. For each strain under study, 100 µl of these suspensions were inoculated into 11 wells containing 100 µl of YPD with different concentrations of Cyclosporine A (CsA) at twice the final concentration [0.39–50 µM]. The 96-well plate was incubated for 24 h at 37 °C, after which the wells were resuspended and the growth of each strain was estimated as a function of OD600. The percentage of inhibition was calculated from the growth in the control well without compound.

## Intracellular ROS

To assess intracellular levels of reactive oxygen species (ROS), we quantified the fluorescence intensity of 2',7'-dichlorofluorescein, the oxidation product of 2',7'-dichlorofluorescein diacetate (H2DCFDA), by flow cytometry using a protocol similar to do Carmo Silva et al (2015). Five wild-type strains and five FLZ strains derived from them, with AUC in DTT higher than the parentals, were selected for assessing the intracellular levels of ROS. Briefly, overnight liquid cultures were prepared and a fresh culture for each strain was adjusted to an OD of 0.1. These cultures were incubated for 3.5 h to reach the exponential growth phase. After incubation, the cells were collected by centrifugation ($6000 \times g$ 5 min), resuspended at 0.1 OD in PBS without (−) or with (+) 100 µM H2DCFDA, and incubated for 30 min. The cultures were then centrifuged again and resuspended in 1 ml PBS for fluorescence measurement using the Beckman Coulter Gallios flow cytometer in the FL1-H channel. A total of 10,000 cells per strain were counted and the mean & median fluorescence intensity with H2DCFDA was compared versus the fluorescence without H2DCFDA and used as a measure of intracellular ROS levels.

## Competition experiments

Competition experiments were performed using fresh 24 h cultures on YPD agar plates. Fresh cells were resuspended in 1 ml of water and counted three times using a DeNovix CellDrop Automated Cell Counter, with the median count used to prepare a dilution of $2.5 \times 10^5$ cells/ml for each strain tested. The strains were mixed in a 1:1 ratio and plated to count the initial number (T0) of colonies growing in triplicate on YPD plates and on YPD plates containing 1 µg/ml anidulafungin, where only anidulafungin-resistant strain

can grow. The strains were also individually plated to ensure the correct cell number and phenotype.

The 1:1 strain mixture was inoculated in triplicate into a 96-well plate, each well containing 200 µL of the corresponding test conditions (YPD, YPD+NaCl 1.5 M, YPD+CsA 20 µM). The plate was incubated for 24 h at 37 °C. Serial dilutions were plated from each well in YPD and YPD+anidulafungin. After 48 h, colonies were counted and the proportion of each strain was calculated by subtracting the number of colonies grown in anidulafungin (multidrug resistant strain) from the number of colonies grown in YPD (wild type strain + multidrug resistant strain). The initial 50/50 ratio at T0 in YPD was as expected and no wild-type strains grew on anidulafungin medium.

In vivo fitness competition was assessed by infecting *Galleria mellonella* larvae, using procedures previously described by our group (Ksiezopolska et al, 2021). Fresh cells were resuspended in 1 ml of Phosphate Buffered Solution (PBS) and counted three times using a DeNovix CellDrop Automated Cell Counter. The median value was used to prepare a dilution of $2.5 \times 10^8$ cells/ml for both the wild type (WT, CBS138) and multidrug-resistant (MDR, AinF_9F) strains. Based on this dilution, the following experimental conditions were prepared: PBS-only, WT only, MDR only and three independent replicates of a 1:1 WT:MDR mixture. For each condition, three *G. mellonella* larvae of similar size and weight were selected. We used high-quality *G. mellonella* larvae that were reared in our laboratory according to the standardized breeding guidelines described previously (de Jong et al, 2022). After briefly cleaning the injection site with 70% ethanol, 10 µL of each inoculum containing a total of $2.5 \times 10^6$ cells per larva was injected. The larvae were then incubated for 24 h at 37 °C in the dark.

Following incubation, the three larvae from each condition were externally washed with 70% ethanol, followed by sterile PBS. The larvae were placed in screw-cap tubes containing three 3-mm glass beads and homogenized using 3 rounds of shaking for 20 s at 4 m/s in a FastPrep-24 homogenizer (MP Biomedicals). The homogenized tissue was resuspended in 1 ml of PBS. Serial dilutions were plated on YPD + chloramphenicol (100 µg/ml) and YPD + chloramphenicol (100 µg/ml) + anidulafungin (1 µg/ml). After 48 h, no colonies were observed in any plate in the PBS-only control. In the WT-only condition, colonies were detected only on YPD + chloramphenicol plates, with no growth on plates containing anidulafungin. In the MDR-only and 1:1 MIX conditions, colonies grew on both types of media. The final MDR/WT ratio was calculated by dividing the number of colonies that grew on the anidulafungin-containing medium (MDR) by the total number of colonies that grew on the medium without anidulafungin (WT + MDR). Results are expressed as the mean and standard deviation from three independent biological replicates, each derived from three infected larvae.

## Checkerboard synergy assay

Assessment of synergistic effects was performed according to the procedure described in (de-la-Fuente et al, 2023) with minor modifications. Briefly, we used a 96-well plate with increasing concentrations of a drug along the columns and increasing concentrations of the test compound along the rows, forming a checkerboard matrix in an 8 × 12 layout. YPD medium was used,

with fluconazole and NaCl diluted independently. We added 50 µl of YPD with 4-fold antifungal concentration, 100 µl of YPD with 2-fold NaCl concentration, and 50 µl of a 4-fold cell suspension. Fluconazole was added to columns 2–12 at increasing (doubled) concentrations ranging from 0.25 to 256 µg/ml. Rows B-H were added with increasing concentrations of NaCl (0.22, 0.33, 0.49, 0.74, 1.11, 1.67, and 2.50 M). Well, A1 served as a growth control containing no drug or NaCl.

Fresh cells were resuspended in 1 ml of water and counted three times using a DeNovix CellDrop Automated Cell Counter, with the median count used to prepare a $4 \times 10^5$ cells/ml dilution for each strain. A 50 µl aliquot of this cell suspension was inoculated into each well, bringing the final volume to 200 µl with a 1x concentration of all components. Three biological replicates were performed. Plates were incubated for 24 h at 37 °C, cultures were resuspended and growth was measured by OD600. Synergy analyses were then performed using Combenefit software on the Bliss and Loewe models (Di Veroli et al, 2016).

## Adaptability assay

To measure the frequency with which N. glabratus strains are spontaneously able to grow under fluconazole and NaCl stress conditions, $10^7$ cells were plated on YPD, YPD + NaCl (1.25 M), YPD + fluconazole (128 µg/ml), and YPD + NaCl + fluconazole (1.25 M and 128 µg/ml, respectively). Fresh 24 h cultures were resuspended in 1 ml of sterile water and counted three times using a DeNovix CellDrop Automated Cell Counter. The median value was used to prepare a suspension of $10^8$ cells/ml. Then, 100 µl of this suspension were spread onto the different media using sterile glass beads and incubated for 72 h. Plates were photographed after incubation using an Epson Perfection V600 Photo scanner.

Similarly, to estimate the frequency of fluconazole-adapted colonies capable of growing under NaCl stress, serial dilutions were plated to obtain ~500 CFU per plate on YPD and YPD + fluconazole (128 µg/ml). After 72 h incubation, colonies were replica plated onto YPD + NaCl (1.25 M) plates. Images were captured after 72 h as described above.

To evaluate N. glabratus adaptability over time, a standardized assay was developed to assess adaptability—the ability of a strain to adapt—under specific conditions. In this assay (Fig. 3B), $1.25 \times 10^5$ cells were incubated in a 96-well plate containing 200 µl of YPD medium with 10 different concentrations of the drug of interest and incubated at 37 °C for 24 h. After incubation, OD600 was measured and the highest drug concentration at which each strain exhibited more than 50% growth was identified as the Maximum Growth Concentration (MGC). Next, 100 µl of the culture from the MGC well was diluted in 900 µl of YPD, and 50 µl of this dilution was inoculated into a fresh set of wells containing the same 12 drug concentrations. This process was repeated for three days more, with each day's MGC population serving as the starting population for the following day.

At the end of the four-day drug exposure, the final MGC population was plated on YPD solid medium without drug and incubated for 72 h without selective pressure to confirm that the acquired resistance remained stable. The final susceptibility of the adapted strains was then assessed by inoculating $1.25 \times 10^5$ cells with the same drug concentrations, followed by 24 h incubation at 37 °C and measurement of OD600. Adaptability (ADB) was calculated by dividing the MGC of the Adapted strain (MGC_A) by the MGC of the parental strain (MGC_P) (Fig. 3D), yielding an ADB value under the specified conditions, indicated by the number of days in brackets and the drug concentration range (e.g., $\text{ADB}_{(4)}{}_{0.5\text{-}256} = 8$). This value represents the number of times the MGC can double during the course of the experiment.

To assess whether a compound could inhibit adaptability under these conditions, the experiment was performed with the experimental compound at a fixed concentration in all wells. The ADB value was then calculated for both the drug alone and the drug in combination with the compound. The Inhibition of Adaptability value (IADB) was obtained by dividing the ADB of the drug alone by the ADB of the drug plus compound (Fig. 3D). These experiments were performed with 10 concentrations of fluconazole ranging from 0.5 to 256 µg/ml either alone or in combination with NaCl (1.25 M). Wild-type strains CBS138 and BG2 were used for these evaluations.

## Modeling the sources of variation in trade-off intensity

We modeled the variation in trade-off intensity (TI, which is $\text{AUC}_R$ for YPD, $\text{fOD}_R$ for CsA and $\text{fAUC}_R$ for other conditions) from two types of predictive features: (i) variants and aneuploidies acquired during in vitro drug adaptation, and (ii) strain background features that were expected to underlie TI changes. The following sections provide further methodological details.

### Software environment
All the data processing and modeling was performed with python scripting 3.9.7 using the following dependencies: scipy v1.11.3 (Virtanen et al, 2020), numpy 1.26.0 (Harris et al, 2020), biopython 1.78 (Cock et al, 2009), pandas 2.1.1 (McKinney, 2010), statsmodels 0.14.0 (Seabold and Perktold, 2010) and scikit-learn 1.3.0 (Pedregosa et al, 2011). All related plots were generated with matplotlib 3.8.0 (Hunter, 2007) and seaborn 0.13.2 (Waskom, 2021). All the code used can be found in the GitHub repository https://github.com/Gabaldonlab/Cglabrata_tradeoffs. The file 'Cglabrata_tradeoffs.yml' in this repository lists all software dependencies.

### Definition and engineering of predictive features
We obtained all variants acquired during in vitro drug adaptation from Ksiezopolska et al (2021), listed in the table 'data/ Ksiezopolska2021_all_new_variants_per_sample_WGS.tab' within our GitHub repository (https://github.com/Gabaldonlab/ Cglabrata_tradeoffs). These include chromosomal aneuploidies (duplications), SNPs, small INDELs, structural and copy-number variants. To keep reliable SNPs and INDELs we retained only those that (i) were not affecting genes also mutated in YPD control conditions and (ii) were not affecting the genes CAGL0G05522g or IRA1 (which generated possibly false positive variants, according to manual curation). The presence/absence pattern of each of these newly acquired variants was considered as a binary predictive feature.

To take into account that different variants (e.g., in the same gene) may have equivalent effects (Ksiezopolska et al, 2021) we grouped mutations according to their predicted functional impacts, resulting in new, engineered, predictive features. On the one hand, one of these engineering strategies consisted in grouping variants

that may have similar effects on the 'resistance-associated' genes and chromosomes, that were recurrently (more than once) mutated during drug adaptation (*ERG3, CNE1, ERG4, PDR1, CDR1, EPA13, ERG11, FKS1, FKS2* and chromosome E (Ksiezopolska et al, 2021). We defined one new feature for each of the genes *ERG3, CNE1, ERG4, PDR1, CDR1* and *EPA13*, which may have the values 'no mutation', 'truncation', 'missense' or 'duplication' across strains, depending on the type of mutations. Also, following the observation that different adaptive paths led to functionally distinct combinations of *FKS1* and *FKS2* variants, we defined a 'FKS profile' feature summarizing this combination (e.g., 'missense mutation in *FKS1* and truncation in *FKS2*' would be one value for this feature). Similarly, given the likely interaction between *ERG11* variants and chromosome E aneuploidies (resulting in *ERG11* duplications), we defined the '*ERG11*/chromosome E profile' feature, representing the combination of such variants. For instance, 'missense mutation in *ERG11* and absence of chromosome E duplication' would be one value for this feature. We then processed each of these features using one-hot encoding to obtain binary features, used in modeling. For example, the feature '*ERG3* profile' (with values 'no mutation', 'truncation' or 'missense') resulted in two binary features: '*ERG3* truncation' and '*ERG3* missense'. Note that we obtained the gene names and descriptions from the Candida Genome Database (CGD) chromosomal feature files (Skrzypek et al, 2016), from strain CBS138 (version s02-m07-r35). For reference, we provide the gene information in the file 'data/SchikoraTamarit2024_Candida_-glabrata_gene_features.tab' of our GitHub repository (https://github.com/Gabaldonlab/Cglabrata_tradeoffs).

On the other hand, we grouped variants depending on their effects on protein regions and pathways. Given that different missense variants in the same protein region could have equivalent effects, we grouped such mutations by all possible protein regions in each resistance-associated gene. For instance, if a gene had variants in the protein positions 140, 150, and 180, we would create the features 'missense in p.140-150', 'missense in p.150-180', and 'missense in p.140-180'. Similarly, we created one feature for each position, representing the presence of missense variants in that protein position (e.g., 'missense in p.140'). To avoid feature redundancy, we created new 'region' features only if they had unique presence/absence patterns across strains, distinct from the pattern related to genes and/or single variants. Also, if two region features had equal patterns, we prioritized the shorter regions. Note that we considered inframe deletions, inframe insertions and loss-of-stop mutations also as "missense" for this protein region grouping.

Similarly, to consider that variants in different genes from the same pathway could have similar effects, we generated engineered features based on such pathway-level collapsing of variants. For this, we considered all genes with any acquired mutation during drug adaptation, even if they were mutated only once. We obtained curated Reactome, Gene Ontology (GO) and MetaCyc per-gene annotations from a previous study from our lab (Schikora-Tamarit and Gabaldón, 2024), available in the tables 'data/SchikoraTamarit2024_annotation_Candida_glabrata_<pathway name>.tab' of our GitHub repository (https://github.com/Gabaldonlab/Cglabrata_tradeoffs). We created one feature for each combination of (i) pathway that had >1 affected gene and (ii) variant effect (duplication, missense, truncation). For instance, one such feature may be 'missense variants in pathway GO:0031227, intrinsic

component of ER membrane'. To avoid redundancy we created new 'pathway' features only if they had unique presence/absence patterns across strains, distinct from the pattern related to genes, single variants and/or protein regions. Also, given that several pathways had equal patterns we iteratively created them using two nested loops. The first loop iterated through the type of pathway in this order: (i) GO biological process, (ii) GO molecular function, (iii) GO cellular component, (iv) MetaCyc and (v) Reactome. The second loop iterated through the variant effect in this order: (i) truncation, (ii) duplication, (iii) missense. For each pathway type and variant effect, we only generated a new feature if its presence/absence pattern did not overlap the pattern of another previously-added pathway. If two pathways of the same type had the same pattern we prioritized those with a longer description length.

On another line, given that the samples belong to highly diverse *N. glabratus* strain backgrounds (Carreté et al, 2018), we hypothesized that pre-existing differences in fitness and stress tolerance could impact TI variation. For instance, we speculate that strains with lower stress tolerance in the wild type may have lower stress tolerance (e.g., $AUC_R$). To test this with our modeling we considered four features that may capture this wild type (WT) strain effect. First, the WT AUC in YPD was taken as an indicator of baseline fitness in a rich medium. Second, we considered the WT AUC in the analyzed condition (e.g., NaCl or CsA), which represents the baseline for raw stress tolerance. Third, we took the $WT\_fAUC$ in the analyzed condition as the baseline for relative stress tolerance. Fourth, the actual strain, one-hot encoded to be a binary predictor, was taken as a feature which captures the precise background. The first three features are continuous, whereas the strain is a binary one.

In summary, we considered various strain features (both binary and continuous) that may capture how the variants acquired during drug adaptation and the strain background impact TI variation. To establish a first feature filter, we only kept binary features (i.e., variants, groups of variants and one-hot encoded strain information) present in >1 in vitro evolution lineages (e.g., strains EB0911_3B_ANI and EB0911_3B_AinF would be from the same lineage, see Ksiezopolska et al (2021)). This resulted in 770 features.

### Univariate modeling of trade-off intensity

To model TI in a univariate way we predicted $AUC_R$, $fOD_R$ or $fAUC_R$ from each of the 770 features independently. To estimate the strength of the association we considered the proportion of variation ($r^2$) yielded by a linear regression model based on each feature. Specifically, to avoid overfit $r^2$ estimates we used the function cross_val_score from the sklearn.model_selection library. This is a pipeline using four-fold cross-validation (CV) to get four linear regression models (based on the implementation from sklearn.linear_model.LinearRegression) per feature, each trained on 75% of the data and tested on the remaining 25%. To ensure that the CV folds were truly independent we customized them, randomly distributing samples by in vitro evolution lineages across folds, ensuring that all samples from the same lineage (e.g., strains EB0911_3B_ANI and EB0911_3B_AinF) are in the same fold. Thus, we obtained four $r^2$ estimates coming from each testing instance, and we used their average as the proportion of variation explained by each feature independently.

Also, we calculated *p* values indicating the significance of the association between each feature and TI. For binary features we

calculated the *p* value of a two-sided Kolmogorov–Smirnov (KS) test comparing the TI distributions of the strains with or without the feature, using scipy.stats.ks_2samp (with the "auto" method). For continuous features we calculated the *p* value of a spearman correlation test between the feature and TI, using the scipy.stats.-spearmanr function. To address the multiple testing bias in each condition we used the module statsmodels.stats.multitest.multi-pletests to obtain Bonferroni-corrected *p* values.

### Multivariate modeling of trade-off intensity

To build multivariate models considering all features simulta-neously we used regression Machine Learning (ML) methods to predict TI variation from a subset of relevant features. This approach was not trivial due to three main reasons. First, there was a high potential for overfitting given the high numbers of predictors. Second, our dataset was relatively small, as for most conditions only a few strains had severe trade-offs (Fig. 5A), which complicated our ability to build accurate and generalizable models. For instance, the typical approach of building the models in ~75% of the data (training set) and testing them on the remaining 25% (testing set) (Vabalas et al, 2019) was deemed unreasonable, as it would lead to only very few samples used for testing. Third, we expected high multicollinearity among predictors, which may lead to a feature selection process that is not robust, but rather highly-dependent on the specific train/test splits used.

To address these limitations we implemented a pipeline to obtain, for each condition, a set of features that predict TI variation (i) in a robust manner (e.g., across multiple random train/test splits of the data) and (ii) in a way that is unexpected by chance (e.g., when predicting randomized TI values). This is expected to address the overfitting and multicollinearity issues, even when using the whole dataset for model building. Note that we used ML models to gain biological insights into the drivers of TI variation, rather than to get highly accurate predictions of TI variation (e.g., with r² > 0.9). For each condition, we build a different model for (i) various statistical methods (multiple linear regression, random forest regression or regression trees), (ii) different feature selection models (e.g., forward selection or recursive feature elimination) and (iii) distinct settings for the models (e.g., tree pruning parameters or the consideration of two-way interactions in linear models). In total, we built 58 models per condition, each of them based on a minimal set of predictive features of TI variation, and yielding a particular predictive power measured with r². In the following sections we provide details on how we performed each of these steps.

**Model building: feature selection and hyperparameter tuning**: To ensure accurate modeling, and avoid being restricted by the usage of a single statistical approach (e.g., linear regression), we applied several types of regression models to our data, namely:

- Linear regression without interactions, which is a simple multi-variate approach in which the target variable is predicted from a linear combination of predictive features, without considering interactions. To run this we used the implementation from sklearn.linear_model.LinearRegression with default parameters.
- Linear regression with interactions, an equivalent method to the previous one but considering all possible two-way interactions between binary features. This allowed us to model interactions within the simple and interpretable frame of linear regression.

- Decision tree regression, a non-parametric approach based on a decision tree that predicts continuous values, like TI. This is a powerful approach to take into account high-order interactions and non-linear effects, which cannot be properly modeled by linear regression. We use the implementation from sklearn.tree.Deci-sionTreeRegressor, with parameters min_samples_split=2 and min_samples_leaf=2.
- Decision tree regression with automatic setting of the Cost-Complexity Pruning (CCP) α parameter. We used this approach to circumvent the typical tendency towards overfitting of regression tree models. CCP is a recommended strategy to build non-overfit trees, which have an adequate level of leaf pruning. It is based on the CCP α parameter, which may adopt a range of values specific to each dataset, and determines how much the tree is pruned (Pedregosa et al, 2011). For any given dataset, we selected the optimal α value our of those yielded by the function sklearn.-tree.DecisionTreeRegressor.cost_complexity_pruning_path. For each α value, and across five independent random CV splits, we calculated the CV-based average r² (as described in the section "Univariate modeling of trade-off intensity"). The selected optimal α was the maximum value that had highest average r² in a majority of random CV splits. To implement this, we configured sklearn.tree.DecisionTreeRegressor with parameter ccp_alpha set to the optimal α.
- Random forest regression, a complex ML method based on generating an ensemble of regression trees, each based on a resampled subset of the training data. We used this approach to harness the suitability of tree-based predictions, while reducing the potential for overfitting. To run such models we used the implementation from sklearn.ensemble.RandomForestRegressor with parameters n_estimators=100, bootstrap=True, min_sam-ples_split=2 and min_samples_leaf=2.

We used each of these models on each condition to find a minimal set of predictive features that predict a reasonable amount of TI variation. For this, we used different feature selection approaches:

- Forward sequential selection, a commonly-used greedy approach, in which features yielding a maximum average r² across CV folds are added in a stepwise manner, until the increase in r² is below a 'tol' parameter (described below). We used the implementation from sklearn.feature_selection.SequentialFeatureSelector, with para-meters n_features_to_select = "auto", direction = "forward" and scoring = "r2". Also, note that we passed a custom CV split through the 'cv' parameter to have lineage-balanced folds (described above). We applied this feature selection method for the linear regression, decision tree regression and random forest regression models. We did not apply to the regression trees with automatic CCP α setting because this function requires an unfitted model, which is incompatible with this model type.
- Custom forward sequential selection, equivalent to the previous one, but implemented by us from scratch to accommodate the automatic CCP α setting for each iteration. We used it for the linear regression, decision tree regression and decision tree regression with automatic CCP α setting models. We did not use it on the random forest-based modeling due to computational constraints.
- Recursive feature elimination with CV, a method that starts with all features and keeps removing features recursively until performance

(measured as $r^2$ across CV folds) is maximized. We used the implementation from sklearn.feature_selection.RFECV, run with parameters step=1, scoring = "r2" and custom CV folds as above (parameter 'cv'). We applied it for decision tree regression and random forest regression models. We did not apply it to the regression trees with automatic CCP α setting due to the same reasons as the SequentialFeatureSelector model. Also, we did not apply it to linear models due to computational constraints.

- Feature importance, in which we fit the model to the whole dataset and extract the top 10 features according to the feature_importances_ attribute of the fitted model. We applied this for all tree-based models, including the random forest regression, but not on the linear regression because it was not possible.
- Select from model selection, a similar approach to the previous ones, in which the features that have a feature importance score above the mean of all features are kept. We used the implementation from sklearn.feature_selection.SelectFromModel with default parameters. We applied this approach for all tree-based models, including the random forest regression. However, we did not apply it to the linear regression algorithms due to the absence of feature importance measurements.

For feature selection we considered two relevant parameters. First, for all model types except random forest regressors we selected features with or without filtering out predictors that had a significant univariate association with TI variation (raw $p < 0.05$, calculated as described in the section "Univariate modeling of trade-off intensity"). Note that, due to computational constraints, we only built linear models considering two-way interactions based on features that had such a univariate association with TI variation. Second, for forward feature selection we tried different 'tol' parameters (0.1, 0.15, and 0.2) to modulate the potential for overfitting.

In summary, we ran a feature selection pipeline for each combination of (i) model type, (ii) feature selection method, (iii) filtering (or not) of features with univariate association and (iv) 'tol' values. In total, we built 58 models per condition, each yielding a minimal set of features. Also, we used the function cross_val_score as defined above to evaluate the average $r^2$ across independent CV folds (as described above), which provides an indicator of model performance. Note that we did not perform additional hyperparameter tuning because we were mostly interested in analyzing the predictive features, rather than optimizing performance. All in all, we generated a collection of predictive features and model performances that constituted an interesting dataset to analyze the determinants of TI variation.

**Calculation of model consistency**: We observed that computing a given model with different random CV splits would often yield different results (i.e., distinct sets of predictive features and performances), suggesting that some of the feature selection runs could lead to misleading conclusions regarding the mechanisms underlying TI variation. To address this we calculated a 'model consistency' score, useful to reduce overfitting, address multicollinearity and pinpoint the best models. To calculate this score we computed each feature selection pipeline (with each model type, selection method, filtering of features with univariate associations and 'tol' value) 10 times using different random CV splits. Then, for each of these 10 CV splits we calculated the model consistency as the number of other splits that yielded the same feature set,

resulting in a number between 1 (very inconsistent model) and 10 (very consistent model). Similarly, we calculated the model performance as the mean average $r^2$ across splits with the same set of features. Finally, for each 58 model/feature selection settings we kept the CV split with the highest consistency and model performance across splits. These metrics were further used to select the best models for each condition (see main text). For instance, we consider that only models with high consistency and high performance across splits are trustworthy and may be used to extract useful biological insights.

**Calculation of maxT p values to identify good models**: To control for overfitting, we inferred the empirical probability of observing equivalent models by chance (with equal or higher $r^2$ and model consistency), using a method inspired by the maxT GWAS approach (Gundlach et al, 2016; Schikora-Tamarit and Gabaldón, 2024). This is an empiric method that yields $p$ values adjusted for multiple testing. To get maxT $p$ values for each of the 58 models/feature selection settings we ran a pipeline equivalent to the one described in the sections above for the real TI data, but trying to predict TI values generated by resampling without replacement the original ones. We ran this modeling of randomized TI values 100 times, obtaining a consistency score and average performance ($r^2$) for each, and then calculating the $p$ values as:

$$p\text{max}T = \left( \sum_{i=1}^{100} 1\,if\left( \max(r^2)_{i,c} \geq r^2 \right) \right)/100$$

Where $max(r^2)_{i,c}$ is the maximum $r^2$ obtained in resample $i$ for models/feature selection settings that yield a model consistency score ($c$) $\geq$ above the score observed when using that same setting on the real TI data. Thus for each setting we got a (i) set of selected features, (ii) model performance ($r^2$), (iii) consistency score and (iv) maxT $p$ value. To pinpoint the most relevant features we focused on 'good models', defined as those that had a maxT $p < 0.05$, consistency $\geq 3$ and average $r^2$ within the 95 percentile of all models. These are represented in Fig. 5C,D.

# Calculating statistical trends on trade-off intensities

To evaluate whether strains evolved in certain drugs (e.g., anidulafungin or fluconazole) showed particular trade-offs on average, we compared their distribution of trade-off intensities (median $fAUC_R$ for stress conditions, $AUC_R$ for YPD and $fOD_R$ for CsA) to 1 (no trade-off). We used a two-sided Wilcoxon test (using scipy.stats.wilcoxon) to ascertain if the Similarly, to define that a strain (e.g., P35_10E_FLZ) had a "significant trade-off" in a given condition (e.g., CFW) it needed two conditions. First, the mean trade-off intensity across technical replicates was below 1, with a $p < 0.05$ according to a one-sided t-test (using scipy.stats.ttest_1-samp). Second, the median fAUCR/AUCR/fODR was <0.9 (significant trade-off) or <0.5 (severe trade-off). This information is found in Dataset EV4. Mean of the distribution for a given condition and evolved drug was different from 1. The results of these tests are found in Dataset EV1.

Similarly, to define that a strain (e.g., P35_10E_FLZ) had a "significant trade-off" in a given condition (e.g., CFW) it needed

two conditions. First, the mean trade-off intensity across technical replicates was below 1, with a $p < 0.05$ according to a one-sided t-test (using scipy.stats.ttest_1samp). Second, the median $fAUC_R/AUC_R/fOD_R$ was <0.9 (significant trade-off) or <0.5 (severe trade-off). This information is found in Dataset EV4.

Due to the objective nature of the quantitative in vitro experiments, no blinding was performed.

## Data availability

The main datasets supporting the findings of this study are included in the Datasets EV1–5, which also contain the results of the statistical analyses. The data sources for all figures are available in individual .zip files. The computer code produced in this study is available via the GitHub repository: https://github.com/Gabaldonlab/Cglabrata_tradeoffs.

The source data of this paper are collected in the following database record: biostudies:S-SCDT-10_1038-S44320-025-00185-3.

## Peer review information

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

## Acknowledgements

The TG group acknowledges support from the Spanish Ministry of Science and Innovation for grants PID2021-126067NB-I00, CPP2021-008552, PCI2022-135066-2, and PDC2022-133266-I00, cofounded by ERDF "A way of making Europe"; from the Catalan Research Agency (AGAUR) SGR01551; from the European Union's Horizon 2020 research and innovation programme (ERC-2016-724173); from the Gordon and Betty Moore Foundation (Grant GBMF9742); from the "La Caixa" foundation (Grant LCF/PR/HR21/00737), and from the Instituto de Salud Carlos III (IMPACT Grant IMP/00019 and CIBERINFEC CB21/13/00061-ISCIII-SGEFI/ERDF). JCNR received a Predoctoral Fellowship from the Spanish Ministry of Science and Innovation (grant number PRE2019-088193). MAS-T received a Predoctoral Fellowship from the 'La Caixa' Foundation (grant number LCF/BQ/DR19/11740023). Some of the figures were created with BioRender.com. The authors wish to thank Dominique Sanglard for kindly sharing DSY562 and DSY565 strains. We thank the Scientific and Technological Centers (CCiTUB), Universitat de Barcelona, and Jaume Comas-Rius for their support and advice on flow cytometry technique. The authors thank all of the members of the Gabaldón group for key support during this work. In particular, E. Ksiezopolska, E. Saus, M. Bernabeu, who provided useful feedback, which was key for the project development.

## Author contributions

**Juan Carlos Nunez-Rodriguez**: Conceptualization; Data curation; Investigation; Visualization; Methodology; Writing—original draft; Writing—review and editing. **Miquel Àngel Schikora-Tamarit**: Data curation; Software; Formal analysis; Visualization; Methodology; Writing—original draft. **Toni Gabaldón**: Conceptualization; Resources; Supervision; Funding acquisition; Validation; Investigation; Writing—original draft; Project administration; Writing—review and editing.

Source data underlying figure panels in this paper may have individual authorship assigned. Where available, figure panel/source data authorship is listed in the following database record: biostudies:S-SCDT-10_1038-S44320-025-00185-3.

## Disclosure and competing interests statement

The authors declare no competing interests.

# Expanded View Figures

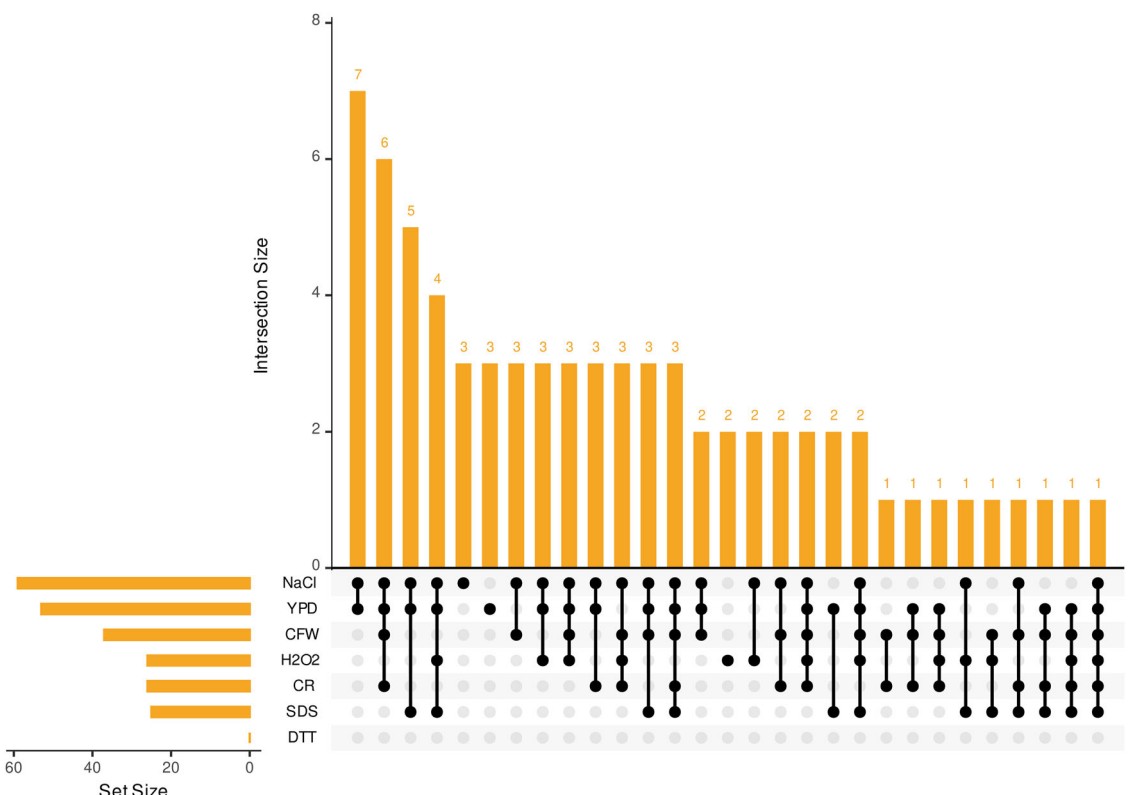

**Figure EV1.  Trade-off combinations observed in collection phenotyping.**

Upset plot showing statistically significant negative trade-off (median fAUCR < 0.9, one-sided t-test (compared to 1) *p* < 0.05, Dataset EV4). Source data are available online for this figure.

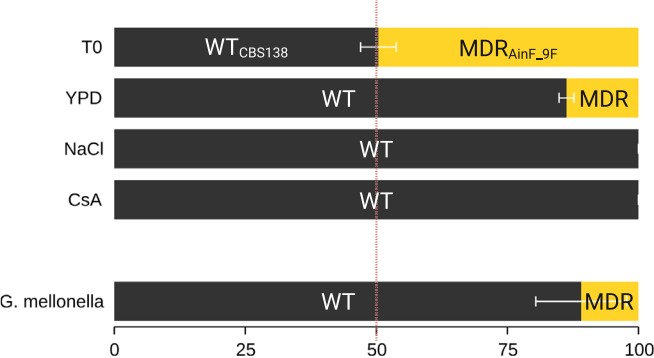

**Figure EV2.  Fitness trade-offs affect competitiveness of drug-resistant strains.**

Results of a competition experiment of the wild type reference strain CBS138 and an multidrug resistant (MDR) strain (AinF-9F) evolved from it. Stacked bar plots showing the percentage of colonies counted after 48 h YPDa plates for each strain at the initial mix of containing 50% of each strain (T0), after 24 h growing in YPD (YPD), in YPD + 1.5 mM NaCl (NaCl), YPD + 20 µM CsA (CsA) and inside the invertebrate model of fungal infection *Galleria mellonella* (*G. mellonella*). Source data are available online for this figure.

    

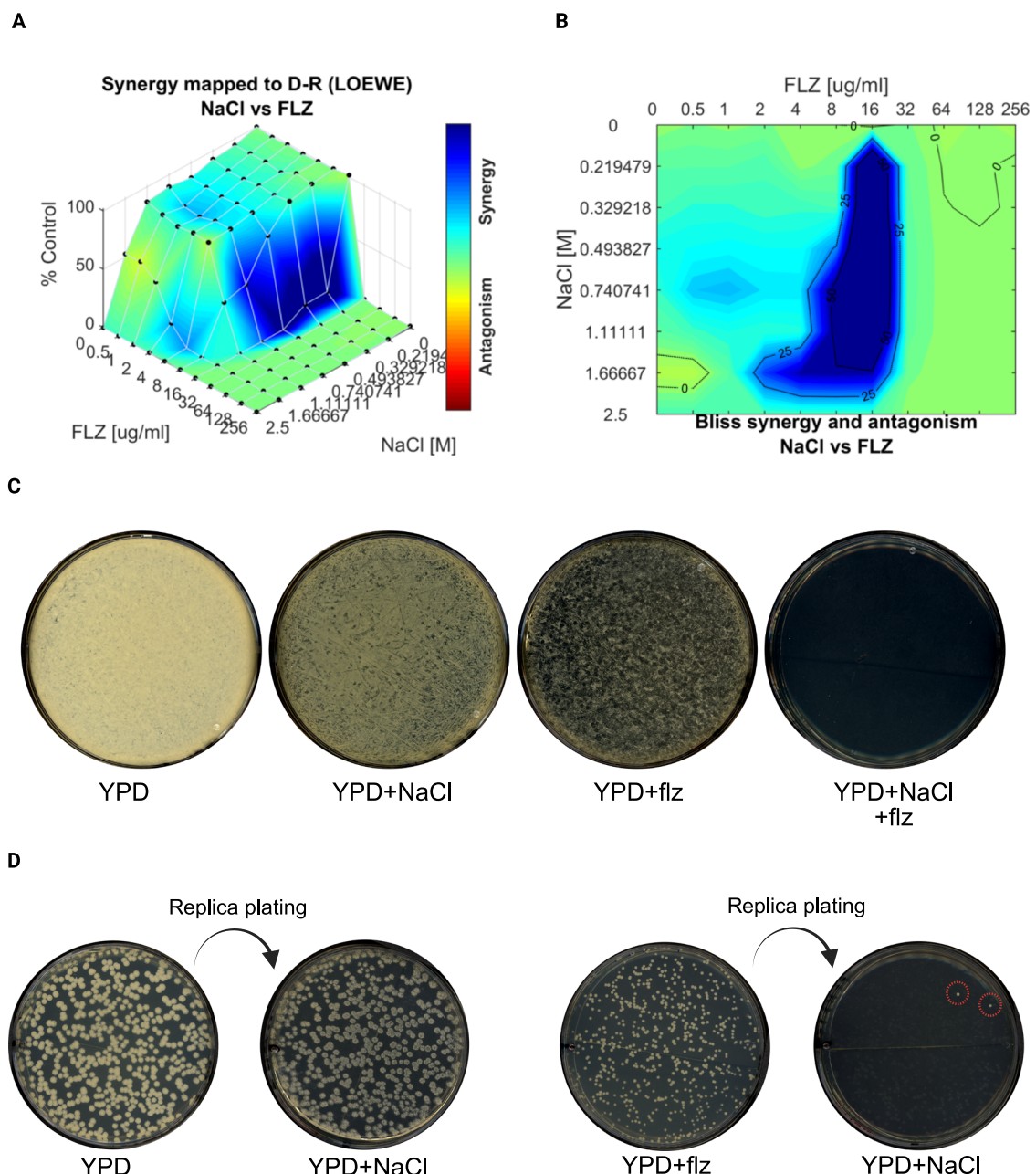

**Figure EV3. Synergistic effect of NaCl with fluconazole on *N. glabratus* growth inhibition.**

(A) 3D representation of the synergy matrix between different concentrations of fluconazole (flz) and NaCl using the Loewe synergy model. (B) Contour plot showing the synergy values calculated from the Bliss model. Both plots were generated by Combenefit software. (C) Images taken after 72 h of incubation of $10^7$ cells on YPD, YPD + 1.25 M NaCl, YPD + 128 μg/mL flz, and YPD + 1.25 M NaCl + 128 μg/mL flz plates; no colonies were observed under the combined treatment. (D) Images of plates obtained after 72 h of replica plating onto YPD + 1.25 M NaCl (shown after the arrow) from source plates (shown before the arrow) containing ~500 colonies of YPD adapted or YPD + flz (128 μg/mL) adapted strains. On the YPD + NaCl plate replicated from YPD + flz, only two colonies were observed, marked with red circles. Source data are available online for this figure.

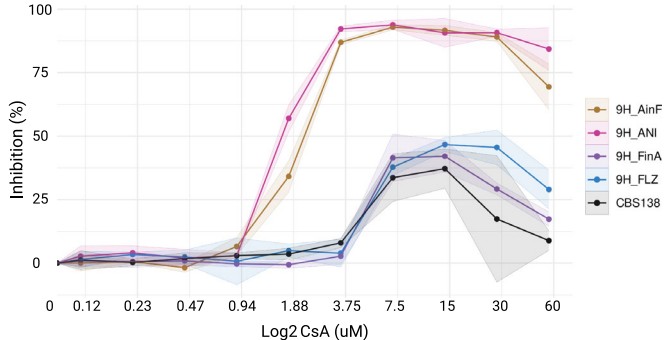

**Figure EV4.   Selective inhibition of Cyclosporine A to certain drug resistant strains.**

Dose–response assay for Cyclosporine A (CsA) in evolved strains under different treatment of the CBS138 wild type reference strain. The y-axis indicates the percentage of growth inhibition versus control based on OD600 measurements, and the x-axis indicates CsA concentrations on a logarithmic scale. Source data are available online for this figure.

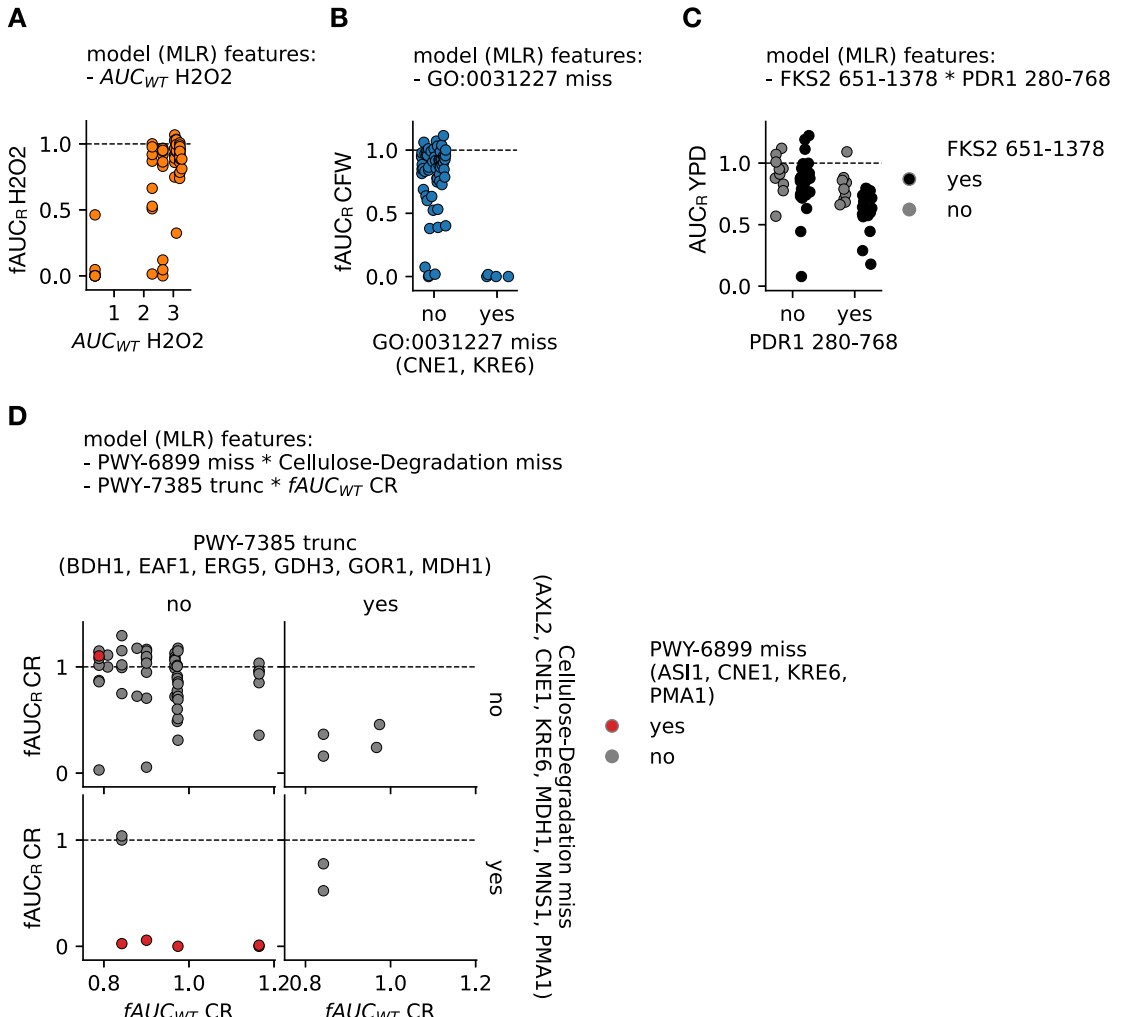

**Figure EV5. Top model results for phenotypes with lower predictive accuracy.**

Panels **A–D** are scatterplots equivalent to those in Fig. 5E–G, but for $H_2O_2$, CFW, YPD, and CR, respectively See Fig. 5D for reference about the pathway-related figures. Source data are available online for this figure.

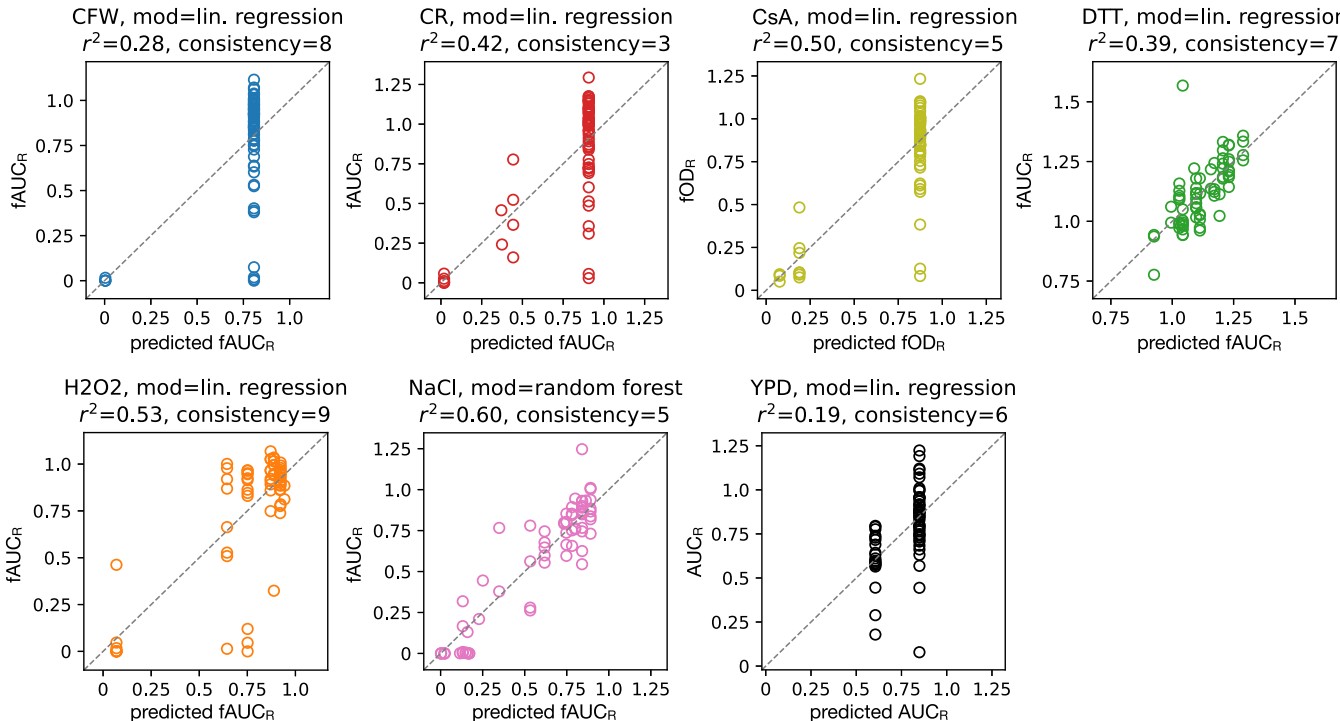

**Figure EV6. Predictive performance of the top models.**

Across all different strains, in each condition, the predicted trade-off intensity (x-axis) vs the actual one (y-axis) for the top models (see Fig. 5C). The title indicates the condition, type of model used (linear regression or random forest), the model performance ($r^2$) and the model consistency. Source data are available online for this figure.

