## [Peer Review File · Molecular Systems Biology]

Uncovering actionable trade-offs of antifungal resistance in a yeast pathogen

Juan Nunez-Rodriguez, Miquel Schikora-Tamarit, and Toni Gabaldón

Corresponding author(s): Toni Gabaldón (toni.gabaldon@bsc.es)

Review Timeline:

Submission Date:	23rd May 25
Editorial Decision:	7th Aug 25
Revision Received:	22nd Oct 25
Editorial Decision:	11th Dec 25
Revision Received:	13th Dec 25
Accepted:	15th Dec 25

Editor: Poonam Bheda

Transaction Report: The first round of review of this manuscript was performed at another journal.

Reviewer Expertise:

Referee #1: Fungal pathogens, phenotypic screens

Referee #2: Fungal pathogens, AMR

Referee #3: Antifungal resistance and therapy

Referee #4: Mathematical modelling

Reviewers Comments:

Reviewer #1 (Remarks to the Author):

This study examined the fitness trade-offs linked to azole and echinocandin resistance in *Nakaseomyces glabratus* (formerly *Candida glabrata*), a yeast pathogen increasingly resistant to antifungal drugs. The researchers profiled a large collection of resistant strains, revealing that 98% of them displayed reduced fitness when exposed to various environmental stresses. Despite differences in genetic backgrounds and resistance mechanisms, they found consistent vulnerabilities associated with resistance. Multivariate modeling showed complex genetic interactions driving these trade-offs. To experimentally support their hypothesis, the authors demonstrated that Cyclosporin A selectively inhibited echinocandin-resistant strains, and NaCl prevented the development of fluconazole resistance. In conclusion, the authors highlighted the potential of targeting resistance-related fitness costs as a strategy to address antifungal resistance.

This study offers valuable insights into the complex interplay between antifungal drug resistance and stress-induced fitness trade-offs in fungal pathogens. The manuscript is well-written and logically structured. However, I have the following major and minor comments for consideration.

[Major comments]

The primary weakness of this study is the absence of *in vivo* data to support the authors' claims. Can the authors observe similar trade-off effects in drug-resistant strains evolved under *in vivo* conditions (e.g., in mice treated with antifungal drugs)? Additionally, if these resistant strains exhibit fitness trade-offs compared to their parental strain, their infectivity and/or virulence might be reduced in a susceptible host in the absence of drug treatment. Based on the current data, it is unclear to what extent these trade-offs affect the stress resistance and survival of *N. glabrata* within a host. While I acknowledge that such experiments may be practically challenging, the lack of *in vivo* supporting data undermines the overall significance of this study.

Response: We agree on the potential interest of adding *in vivo* data. However, this work includes a thorough experimental characterisation of the trade-offs in a large collection of resistant strains, as the main focus of our research was to understand the diversity and genetic basis of potential trade-offs related to drug resistance. We consider it impractical to perform *in vivo* experiments of the same scale, due to practical, economical, and ethical constraints. Experimental vs *in-vivo* work have different pros and cons and we think both are

complementary. We do think our work does provide results that are significant. Nevertheless to explore some of the questions raised by the reviewer we have now tested two serial isolates representing one acquisition of resistance in a patient, and show that the resistant strain shows the NaCl trade-off, underscoring that our findings may reflect processes that can happen in a real clinical situation. Moreover, we have used a Galleria model and one of the strains to validate some of our results in-vivo. Furthermore, we believe that the study of trade-off emergence is inherently valuable, as it illuminates the fundamental evolutionary processes at play and informs strategies to combat drug resistance. While certain trade-offs might not directly translate into a competitive disadvantage during in vivo infection (e.g., in humans or mice), their identification is still highly relevant for the development of novel antifungal strategies. Indeed, trade-offs that do not impose a fitness cost during infection are more likely to become established within a population, making them particularly attractive targets for interventions against resistant strains. We have added these clarifications to the discussion section.

The authors state that fitness trade-offs are a common feature of drug adaptation (lines 71–72). However, this claim may be overstated, as they did not investigate drug resistance patterns in other fungal pathogens. I am curious whether this phenomenon also occurs in other species, such as *Candida albicans*.

Response: We acknowledge that it would be valuable to explore this phenomenon in other species. Nevertheless, the core of our study lies in providing an in-depth understanding of what occurs within the considerable diversity of *Nakaseomyces glabratus*. We've made sure to emphasize this species-specific focus throughout the text.

The authors reported that 98% of drug-resistant strains displayed reduced fitness under at least one stress condition compared to their susceptible parent strains. However, I am curious what percentage of drug-resistant strains demonstrated increased fitness under at least one stress condition.

Response: We've now included this information in the updated Supplemental Table 4. This table shows the list of strains that exhibited statistically significant increases in fitness, using the same thresholds applied for the negative trade-offs.

The authors suggested that co-targeting the osmotic stress response pathway could prevent the emergence of fluconazole resistance based on the effect of NaCl (lines 841–842). However, they should address the possibility that this strategy may not be universally applicable across all fungal pathogens. For instance, inhibition of the HOG pathway, which regulates osmotic stress response, increases fluconazole resistance in *Cryptococcus neoformans* but enhances fluconazole susceptibility in *Candida albicans*.

Response: We agree that these results are specific for *Nakaseomyces glabratus*, and we have emphasized this fact throughout the text.

[Minor comments]

Line 45: "1,5 million" should read "1.5 million".

Response: corrected

Figure 2A and 2C need a statistical analysis. Otherwise, it is difficult to judge whether the differences are significant or not.

Response: The statistical analysis is provided in Supplemental Table 1. As suggested by the reviewer, we've now also added the corresponding asterisks to Figures 2A and 2C to visually indicate significance.

Reference 7 is not an appropriate citation. Please deposit this submitted article to bioRxiv or another public pre-print database.

Response: Reference 7 was a temporary citation for a manuscript that was in press. We're happy to confirm that the article has now been officially published in Nature Protocols. The updated citation will be included in the next revised version of our manuscript:

Nunez-Rodriguez, J.C., Schikora-Tamarit, M.À., Ksiezopolska, Gabaldón, T. Simple large-scale quantitative phenotyping and antimicrobial susceptibility testing with Q-PHAST. Nature Protocol (2025). <https://doi.org/10.1038/s41596-025-01179-z>

In the Methods section, the authors need to pay more attention to using consistent units. For example, both 'uL' and 'ul' (or hours and h) are being currently used

Response: corrected

References should be properly formatted.

Response: We have now re-formatted all citations using appropriate citation management software.

Reviewer #2 (Remarks to the Author):

This manuscript aims to explore fitness trade-offs associated with azole and echinocandin resistance in *Nakaseomyces glabratus* (*Candida glabrata*) and suggests that these trade-offs could be exploited therapeutically. While the topic is highly relevant given the clinical challenges posed by multidrug-resistant fungal pathogens, the study suffers from significant conceptual, methodological, and interpretative weaknesses that undermine its overall conclusions. The study falls short of delivering genuinely actionable insights. The therapeutic potential of targeting trade-offs remains speculative, and the reliance on *in vitro* models without translational validation diminishes the clinical significance. Therefore, the manuscript does not meet the standards for publication in Nature Microbiology.

Major Concerns:

- The idea that antimicrobial resistance carries a fitness trade-off is well-established in bacterial and fungal systems. The authors fail to convincingly articulate what fundamentally new insights their study provides beyond previous work.

Response: We agree with the reviewer that fitness trade-offs are a well-established concept across various biological systems. However, we believe there's a significant knowledge gap regarding their generality and prevalence specifically within resistant to drugs in fungal pathogens.

Furthermore, our research offers several fundamentally new insights. We conducted experiments on a large collection under different conditions and replicates, generating a large and highly relevant dataset on trade-offs in *N. glabratus*. This extensive characterization provides a robust foundation for understanding these phenomena. We applied innovative machine learning models to our data that allowed us to identify potential underlying causes and mechanisms driving these trade-offs, moving beyond mere observation to mechanistic insights in this highly complex type of interactions. Crucially, we also demonstrate the concept that studying these trade-offs can directly lead to the proposal of novel intervention strategies against drug-resistant strains. In our opinion, we consider that we have significantly contributed to the knowledge of this field and laid certain foundations so that it can be expanded in the future.

- The manuscript positions itself as uncovering "actionable trade-offs," but the therapeutic potential of the findings is overstated. The proposed interventions (e.g., NaCl or cyclosporin co-treatment) are impractical in clinical settings. Likewise, the effect of cyclosporin A (CsA) on echinocandin-resistant strains is intriguing but not well substantiated, as the authors do not test whether this sensitivity is relevant under host-mimicking or in vivo conditions.

Response: We agree that these specific applications are not directly practical in current clinical settings, as our research is at the basic stage. However, our manuscript aims to provide novel contributions and proof-of-concept demonstrations that could be foundational for future, innovative drug discovery projects. The exploration of actionable trade-offs, even if not immediately translatable, opens new avenues for research into therapies against resistant strains. We believe developing these new therapeutic compounds or strategies is outside the scope of the current article, but our work lays crucial groundwork for it.

- The study relies heavily on in vitro stress assays using a single phenotyping method (Q-PHAST), without validation in clinically relevant models such as in vivo infection systems or co-culture setups that better mimic host environments. The reliance on spot assays in highly controlled conditions may not accurately reflect the true fitness costs in fluctuating host environments.

Response: We agree that the variable conditions within a host environment are not fully reproducible through in vitro assays. However, the standardization and precise measurement of phenotypes under controlled conditions are crucial for accurately capturing the emergence of trade-offs. Without this controlled approach, the high variability inherent in more complex systems would obscure the results.

As we've added to the discussion, we believe that the identification of these trade-offs is highly relevant for potential treatments, irrespective of whether these trade-offs directly impact the ability to infect the host.

- The section on competition experiments (Lines 601-608) and citation of Figure 4B is misplaced within the section on azoles and NaCl, causing confusion.

Response: The competition assay indeed tested various conditions discussed across the paper, including azole and NaCl as discussed in this specific section, but also others presented later in the study. This was the rationale for including it in Figure 4, where the effect of CsA (Cyclosporin A), the last condition, is discussed.

However, we agree that this placement could cause confusion. We're open to suggestions for relocating it to another main figure or moving it to a dedicated supplementary figure if that would improve clarity.

- The genetic analyses attempt to identify drivers of trade-offs, but the statistical approach is questionable. The multivariate modeling lacks rigorous validation, and the inclusion of over 770 features in predictive modeling raises concerns about overfitting.

Response: We agree that the manuscript lacked sufficient detail to justify how we addressed the overfitting issue. Although it is true that we have many more features than samples, potentially leading to overfitting if using traditional machine learning (ML) approaches, we used various techniques to mitigate this risk. First, we used model consistency scores to ensure that the feature selection process converged to similar results on different cross validation splits. Second, such splits were specifically designed to always include different independently evolved lineages in the training / testing. Third, we used maxT p values to ensure that the observed r^2 and consistency of the final models were higher than expected by chance, providing a direct statistical framework to avoid overfitting. We have clarified these points in the manuscript, highlighting that all these tools were necessary to minimize overfitting.

Moreover, the model does not introduce novel insights, as its findings are not experimentally validated, significantly diminishing its impact.

Response: We agree that further experiments would be necessary to fully validate the hypotheses generated by the models. However, given the large number of hypotheses generated and the dependency on the strain genetic background, we consider that validating these to be a long term project that is outside the scope of the current manuscript. Moreover, we disagree on the general observation that computational models do not offer novel insights. They were key to provide hypotheses on the specific mutations driving variation in tradeoff intensity towards various stressors (particularly DTT, NaCl and CsA). Given the multivariate nature of such genotype-phenotype relationships, we consider that our approach was appropriate for the task. We now stress that the purpose of the models is to derive testable hypotheses on the implication of certain genes in particular phenotypes.

- The claim that 98% of resistant strains exhibit fitness trade-offs is misleading. A minor reduction in growth under a single stress condition does not necessarily constitute a biologically/clinically meaningful trade-off.

Response: We appreciate this important point and agree that a statistically significant reduction in growth under a single stress condition doesn't automatically translate to a clinically meaningful trade-off.

To address this, we've included a specific discussion in the manuscript. We clarify that our definition of a trade-off primarily reflects a consistent evolutionary trend that emerges with resistance. While these trade-offs may not all be directly impactful in a clinical setting, understanding this pervasive evolutionary pattern is crucial because it can be potentially exploited for future therapeutic strategies. Our aim is to highlight these inherent biological compromises as potential vulnerabilities, rather than to claim immediate clinical relevance for every observed instance. If deemed necessary, we can discuss this idea more thoroughly.

- The study fails to explore the mechanistic basis for the synergy between fluconazole and salt stress. It remains unclear whether the observed effect results from stress-induced cellular responses or inhibition of adaptive mechanisms.

Response: We agree that our study doesn't fully resolve the mechanistic basis for the synergy between fluconazole and salt stress. However, we believe this is a sufficiently robust and significant finding that warrants sharing with the scientific community, even if a detailed mechanistic explanation is currently beyond the scope of this paper. These results also add valuable evidence to the growing body of research from other groups exploring the intricate link between osmotic stress responses and drug sensitivity.

- The key role of the calcineurin pathway in echinocandin resistance is well established in *C. glabrata* as acknowledged in the manuscript's conclusions. Given this, the findings related to cyclosporin A do not provide new insights.

Response: We agree that the role of genes related to the calcineurin pathway in echinocandin resistance is well-established, as noted in our conclusions. However, we believe our finding that specific mutations leading to drug resistance can, in turn, trigger a sensitivity to cyclosporin A in resistant strains is a novel insight. While existing literature certainly aids in explaining the underlying mechanism, this specific sensitivity in resistant strains hasn't been previously described. This observation further underscores the potential of trade-offs as a targetable process, which we also present as a new perspective in the field.

- The term "evolvability" is misapplied in this study. The comparison between fluconazole and fluconazole + NaCl is flawed because cells grown with fluconazole alone undergo more divisions than those in fluconazole + NaCl, which do not reach similar OD levels. Consequently, the conclusions regarding "evolvability" should be toned down.

Response: We are open to rephrasing some of our claims to better reflect the nuances of our data. While we acknowledge that cells grown with fluconazole alone undergo more divisions, we still believe our data demonstrates a mechanistic link between fluconazole and NaCl that underlies the lack of resistance acquisition in the presence of NaCl. The

progressive increase in fluconazole resistance in the fluconazole-only condition highlights that resistance acquisition under these conditions is highly frequent, allowing strains to grow in concentrations up to 256 ug/ml within just four passages. A lower number of divisions in the fluconazole + NaCl condition might lead to a softer resistance acquisition curve. However, this difference in division rate isn't significant enough to fully explain the complete absence of resistance acquisition in those strains.

Minor Concerns:

- The introduction lacks a critical discussion of previous work on antifungal resistance trade-offs, particularly studies that have examined fitness costs in *Candida* spp.

Response: For reasons related to the type of editorial style we are addressing, we have provided a brief introduction, but we can expand it if necessary.

- The methods section is overly detailed in some areas (e.g., listing software dependencies) while lacking clarity in key experimental design aspects (e.g., defining "significant" trade-offs).

Response: We consider that listing software dependencies is a necessary level of detail to ensure proper research reproducibility. Note that we used various non-standard approaches for the ML modelling, which may be useful in further studies, so that providing dependencies seems key. We've aimed to maintain this consistent criteria across other aspects of the methods. We have now explicitly defined "significant trade-offs" in both the Methods and Results sections.

Reviewer #3 (Remarks to the Author):

In this manuscript, Nunez-Rodriguez et al. examine a previously established collection of *N. glabratus* isolates evolved in vitro against increasing concentrations of fluconazole or anidulafungin for fitness defects under a panel of environments designed to simulate specific stressors. They report that evolved isolates generally exhibit fitness defects compared to parental strains and similarly show defects when exposed to certain stressors. The authors focus on NaCl as a model of osmotic stress (though they do not argue whether this particular stress is the aspect of growing in NaCl that is causative for their measured phenotypes), which shows broadly increased susceptibility among evolved isolates. NaCl shows a synergistic interaction with fluconazole and the authors present evidence suggesting fluconazole resistance evolution is slowed in the presence of NaCl. The authors then change directions and suggest a global stress response pathway is likely important to sustain resistance, and demonstrate that anidulafungin-evolved isolates show increased susceptibility to cyclosporin A, a calcineurin inhibitor. The authors then model genomic features associated with their stress data. In total, this manuscript adds to a current body of literature suggesting connections between stress responses and antifungal resistance. The major strength of the findings comes from their profiling in a somewhat diverse collection of isolates, though not necessarily from major mechanistic insights associated with these findings. The manuscript could be further strengthened by considering the following specific points:

Major Points:

- The dataset used in Fig. 1 appears to include a number of evolved mutants with nearly 0 growth even under YPD. These are not directly discussed in the text, but it is interesting to recognize that they do not seem to be universally poor growing. For instance, all of the ANI evolved mutants grow well in DTT, presumably including these ones. A) At first glance it's unclear why these strains would be included if they barely grow. Do they grow better when anidulafungin is present in the media? B) Can the authors infer any mechanistic connections in the stressful environments based on when these particular strains are able to grow well?

Response: We acknowledge that this point may initially appear to be confusing. However, the figures illustrate relative growth (AUC_R or fAUC_R), as detailed comprehensively in both the Methods section and Figure 1D. It is important to remark that a relative growth value of 1, indicating similar relative growth (vs rich media) under stress compared to the parental strain (fAUC_R), does not signify a recovery of absolute growth capacity. If the growth in rich medium is inherently low, and this low growth is maintained under a specific stress condition, the resulting fAUC_R will be close to 1. This approach ensures that our analysis accurately reflects specific stress-induced trade-offs rather than overall growth deficiencies.

- Figure 1 The evolved mutants generally have reduced fitness in the absence of any of the stressors used. This seems like it would confound the univariate interpretation of any fitness defects in Fig 1A. Have the authors examined how the fitness profiles differ between the univariate measurements in the current manuscript and perhaps controlling those measurements for the fitness defects in the absence of stress?

Response: As discussed in our previous response and detailed in the Methods section, our analysis relies on fAUC_R calculations precisely to account for these baseline fitness defects. This normalization approach allows us to compare each evolved strain directly to its own parental strain from which it was derived. By doing so, we effectively control for inherent growth differences, ensuring that the fitness defects presented in Figure 1A specifically reflect responses to the applied stressors, rather than confounding them with general fitness costs.

- Figure 3C: I could not determine from the text whether the evolvability assay controls for differences in cell divisions. I might expect cells to be growing more slowly in NaCl (especially in the presence of drug as the authors have pointed out), and reduced cell divisions would be directly explanatory for reduced opportunities to accumulate resistance mutations.

Response: While we agree there might be a difference in the number of cell divisions between the conditions, as we've discussed previously, we don't believe this difference is significant enough to fully explain the complete absence of resistance acquisition when NaCl is present. In line with that, under drug-only conditions, where initial cell divisions are also impacted, the wild-type strains still efficiently increase their resistance. This suggests that while division rate plays a role, the presence of NaCl introduces a more fundamental barrier to resistance evolution, rather than simply reducing the opportunities for mutations.

Minor Points:

- Figure 2C is mentioned in the text before other components of Figure 2. Figure 4 is mentioned before Figure 3. This complicates following the train of logic.

Response: For Figure 2, the current ordering was chosen to best align with the visual structure of the figure's components. Regarding Figure 4B, as we've discussed previously, this panel presents data from a competition assay that encompasses various conditions. These conditions culminate with those explained in the section pertaining to Figure 4. Presenting this data earlier would lack the necessary context established in the preceding sections. Nevertheless, we are open to revising the figure order or placement if the reviewers or editors believe it would enhance the logical flow of the manuscript.

- Lines 610-618, supp Fig 2: The authors argue that a synergistic inhibition between NaCl and Fluconazole suggests the mechanisms driving resistance to these stresses overlap. In Figure 1A, increased NaCl sensitivity is observed universally, so one might expect a similar synergistic response with Anidulafungin as well. Following the author's logic, does this mean there is an overlapping response to NaCl stress and Anidulafungin stress? Between Anidulafungin stress and Fluconazole stress? While this may be true, the authors could strengthen their argument by proposing and perhaps investigating a mechanism by which NaCl would cause an overlapping stress with these compounds, in addition to their assertion that chromosome E duplication is associated with reduced fitness in NaCl.

Response: Our preliminary data investigating the interactions between anidulafungin and NaCl were not conclusive. This was primarily because the rate of anidulafungin resistance acquisition was very low due to its fungicidal, rather than fungistatic, effect. On the other hand, as the reviewer rightly pointed out, we did find a significant correlation between Chromosome E duplications and reduced fitness in NaCl. However, despite analyzing dozens of strains with observed defects, we haven't been able to identify enough conclusive features to propose a specific mechanism explaining this phenomenon. Our results point towards complex epistatic relationships that are highly background-dependent. This inherent complexity currently prevents us from proposing a clear, singular mechanism for these observed effects.

Reviewer #4 (Remarks to the Author):

Nunes-Rodriguez and other authors present an analysis of the tradeoffs associated with evolution of antibiotic resistance in a significant fungal pathogen. A family of antibiotic resistant strains were already developed by the study authors, and the novelty of this study is that they use an apparently new technique, called Q-PHAST to quantify the tradeoffs in the fungus' fitness, specifically in terms of the ability of the antibiotic resistant strains to resist other stressors (e.g. SDS, NaCl, H₂O₂). Substantial statistical modeling is performed to attempt to identify the source of these tradeoffs, focusing on known genes that are associated with antibiotic resistance, and there is some mechanistic discussion of why tradeoffs might emerge.

The study design seems appropriate, and the conclusions reached in the study are sufficiently supported by the evidence that is gathered. There is a large existing literature on the fitness interactions and antibiotic resistance in bacteria (see e.g. the reviews by DI Andersson, and while I was not happy to see citation of this prior work being consigned to a single very high level opinion article (ref. 2), there is less study of eukaryotic antibiotic resistance, and this, plus the focus on stress sensitivity as a fitness cost, and the potential clinical relevance of the work creates a niche for the work in this paper. However, I do not think that niche is Nature Microbiology but rather a more specialized journal for the following reasons:

1. The title, and to some extent the text of the paper, tout that these fitness tradeoffs are actionable. Certainly that is a rich context, and has informed a lot of good work on antibiotic interactions (e.g. from Roy Kishony's group and others), but it is not clear to me that stress sensitivity is a particularly compelling pathway to preventing antibiotic resistance evolution, at least at the level that it will command the interest of Nat. Microbiol's large and diverse readership. There is speculation about calcineurin inhibition being a new therapeutic pathway but I did not find that convincing enough to merit the framing of the paper.

Response: We appreciate the reviewer's valuable feedback regarding the relevance of our study and the potential for these types of investigations to create new research avenues.

We believe this study strongly demonstrates that fitness trade-offs associated with antifungal resistance are common and potentially exploitable. This is a central finding that we believe is robustly supported by our data.

We agree that potential strategies don't solely need to focus on stress sensitivity or the direct use of calcineurin inhibition as a therapeutic approach. Instead, these examples serve as solid proof-of-concept that such trade-offs indeed occur and open the door for exploring them as new potential therapeutic targets. Our aim is to highlight the broader principle that understanding these trade-offs can lead to novel strategies against drug resistance, which we believe will be of interest to a broad scientific community.

2. The work lacks general context in terms of the prior work done on antibiotic resistance and its costs and evolutionary consequences. Even if that is not the focus of the current work, the authors need to be aware that this context is important, particularly for a general interest journal, where a case for broad significance needs to be made by every paper.

Response: We appreciate the comment on the broader context of our work. We initially aimed for a concise format to align with the editorial style of the journal. However, we agree to strengthen the context also in prokaryotes if deemed necessary.

3. Delving more deeply into the results that are presented in the paper, although I found the experiments to be appropriate, and the conclusions well supported, I found weakness in the statistical modeling. I studied this section of the paper with particular care, since the mechanistic interpretations of the data are rooted in this modeling. The authors certainly show some care to deal with the problem that they identify, of having large models, few little data to constrain those models with, and the attendant concerns about overfitting their parameters. I think they took their robustness analysis as far as it can be taken. However, ultimately the overfitting problem is innate to the nature of the fitting they are doing and their

available data. It seems to me that there are testable hypotheses emerging from this part of the paper: e.g.

Response: We see now that the manuscript lacked sufficient detail to justify how we addressed the overfitting issue. Although it is true that we have many more features than samples, potentially leading to overfitting if using traditional machine learning (ML) approaches, we used various techniques to mitigate such a risk. First, we used model consistency scores to ensure that the feature selection process converged to similar results on different cross validation splits. Second, such splits were specifically designed to always include different independently evolved lineages in the training / testing. Third, we used maxT p values to ensure that the observed r^2 and consistency of the final models were higher than expected by chance, providing a direct statistical framework to avoid overfitting. We have clarified these points in the manuscript, highlighting that all these tools were necessary to minimize overfitting. Also, we now stress that, as mentioned by the reviewer, the models are mostly for hypothesis generation, which is a useful output of our approach.

Reviewer #4 (Remarks on code availability):

I reviewed the methods and the results, but I did not think that the modeling results were interesting enough that a detailed code review (e.g. of the hyperparameters) was necessary.

Response: Since we did various non-standard ML steps (e.g. calculating feature consistency and maxT p values) we considered that it was appropriate to explain in detail how we performed the modelling processes. Note that, as detailed in the methods, we did not explore varying model hyperparameters (e.g. from the linear regression or random forest), but rather change model type and feature selection method.

7th Aug 2025

Manuscript Number: MSB-2025-13139-T

Title: Uncovering actionable trade-offs of antifungal resistance in a yeast pathogen

Dear Dr. Gabaldón,

Thank you for the submission of your revised manuscript to Molecular Systems Biology. Unfortunately we were not able to secure the original reviewers of your manuscript, but we have now received the enclosed reports from two new referees that were asked to assess your point-by-point response to the previous reviewers' concerns and revisions to the manuscript it, overlooking the request for additional in vivo relevance. As you will see the new Reviewer 5 finds the quality of the analysis very good, though both Reviewers 5 and 6 mention that the novelty and impact are an ongoing issue (Reviewer 6 conveyed this via a set of questions that we ask reviewers to answer). Reviewer 6 further comments to individual concerns that for the most part said authors responded appropriately to the reviewer comments, with exceptions that additional analysis of strains that might be observed clinically could improve the in vivo relevance and that additional experiments to appropriately compare strains growing at different rates would address the issues of the term "evolvability" that both Reviewers 2 and 3 had mentioned. Although we are willing to overrule a full experimental investigation into the in vivo relevance and find the novelty sufficient for Molecular Systems Biology, we would ask you to address the remaining concerns by Reviewer 6 in a revision. Please note that regarding the comment that the Methods overly detailed in some places and vague in others, we would ask you to not move any Methods to the Supplementary Information, but rather improve the details for any vague descriptions and keep all of the information in the main manuscript as per our policy.

Along with the revision, we would ask you to address the following formatting requirements:

- 1) Please download the EMBO Press "Author Checklist" and complete all relevant questions. This file should be uploaded with your submission. This file can be downloaded from our website at:
<https://www.embopress.org/page/journal/17444292/authorguide>
- 2) Please include keywords to max. 5.
- 3) Please include a Data availability section describing how the data, code etc. have been made available. This section needs to be formatted according to the example below:
"The datasets and computer code produced in this study are available in the following databases:
- Chip-Seq data: Gene Expression Omnibus GSE46748 (<https://www.ncbi.nlm.nih.gov/geo/query/acc.cgi?acc=GSE46748>)
- Modeling computer scripts: GitHub (<https://github.com/SysBioChalmers/GECKO/releases/tag/v1.0>)
- [data type]: [full name of the resource] [accession number/identifier] ([doi or URL or identifiers.org/DATABASE:ACCESSION])"
- 4) Please include a "Disclosure and competing interests statement". We updated our journal's competing interests policy in January 2022 and request authors to consider both actual and perceived competing interests. Please review the policy <https://www.embopress.org/competing-interests> and update your competing interests if necessary.
- 5) Please remove the heading 'Contributions' from the manuscript and specify author contributions in our submission system. CRedit has replaced the traditional author contributions section because it offers a systematic machine-readable author contributions format that allows for more effective research assessment. You are encouraged to use the free text boxes beneath each contributing author's name to add specific details on the author's contribution. More information is available in our guide to authors:
<https://www.embopress.org/page/journal/17574684/authorguide#authorshipguidelines>
- 6) References: Please correct the reference citation in the reference list to be alphabetical (not numerical). Where there are more than 10 authors on a paper, only the first 10 should be listed, followed by "et al.". Please check "Author Guidelines" for more information.
<https://www.embopress.org/page/journal/17574684/authorguide#referencesformat>
- 7) Our journal encourages inclusion of *data citations in the reference list* to directly cite datasets that were re-used and obtained from public databases. Data citations in the article text are distinct from normal bibliographical citations and should directly link to the database records from which the data can be accessed. In the main text, data citations are formatted as follows: "Data ref: Smith et al, 2001" or "Data ref: NCBI Sequence Read Archive PRJNA342805, 2017". In the Reference list, data citations must be labeled with "[DATASET]". A data reference must provide the database name, accession number/identifiers and a resolvable link to the landing page from which the data can be accessed at the end of the reference. Further instructions are available at .
- 8) Please ensure that a statement on whether or not blinding was done is included in the Methods even if no blinding was done. Please also be sure to include the information in the Author Checklist on where it can be found in the manuscript.
- 9) All Materials and Methods need to be described in the main text using our 'Structured Methods' format. According to this format, the Methods section includes a Reagents and Tools Table (listing key reagents, experimental models, software and relevant equipment and including their sources and relevant identifiers) followed by a Methods and Protocols section describing the methods, ideally using a step-by-step protocol format. The aim is to facilitate adoption of the methodologies across labs. Please download and fill our Reagents and Tools Table template (.docx), which you can find in our author guidelines:
<https://www.embopress.org/page/journal/14693178/authorguide#structuredmethods>.

<https://www.embopress.org/doi/10.15252/msb.20178071>. "

10) Please place individual sections of the manuscript in the following order: Title page - Abstract & Keywords - Introduction - Results - Discussion - Methods - Data Availability - Acknowledgements - Disclosure and Competing Interests Statement - References - Figure Legends - Expanded View Figure Legends.

11) For the figures and figure legends, please take care of the following:

- Please remove all figures from main manuscript file and leave only the main figure legends placed after the references.
- All figure and panel callouts should be sequential in the manuscript.
- Supplemental/Supplementary Figure 1-4 should be renamed to Figure EV1-EV4. Each figure will still need to fit onto one page. Please ensure that the callouts are also updated in the main manuscript. The legends should stay in the manuscript, with the heading Expanded View Figures Legends, and placed after the main figure legends.
- Main figures and EV figures should be uploaded as individual, high-resolution files.
- Please provide exact p values in the figures or figure legends of Figure 2A, C; S1.
- Please note that the box plots need to be defined in terms of minima, maxima, bounds of box and whiskers, and percentile in the legends of figures 2A, C

12) Please upload all supplementary tables as one .xsl file per table and rename them to Dataset EV1-5 (filenames and titles within the file). Each dataset will need its legend removed from the manuscript and added to the corresponding file in a separate tab. Please also update their callouts in main manuscript text.

13) Please note that funding information should be given in an "Acknowledgements" section. Please ensure that all funding sources are entered into the manuscript submission system.

14) Synopsis:

- Synopsis image: Please provide a graphic that summarises the main findings of the manuscript on a glance and upload it as a high-resolution jpeg file 550 pixels wide x (300-600) pixels high.
- Synopsis text: Please provide a short standfirst (maximum of 300 characters, including space), limit the bullet points to max. 5 and upload it as a separate .doc file. Please write the bullet points to summarise the key NEW findings. They should be designed to be complementary to the abstract - i.e. not repeat the same text. We encourage inclusion of key acronyms and quantitative information (maximum of 30 words / bullet point). Please use the passive voice.
- Please check your synopsis text and image before submission with your revised manuscript. Please be aware that in the proof stage minor corrections only are allowed (e.g., typos).

15) Source Data: Please ensure that a completed Source Data checklist is uploaded as a Related Manuscript File. You will be contacted separately with instructions with the checklist for the Source Data. Source Data should be organized as a single source data file (zipped) per figure for main figures (all EV and/or Appendix figure Source Data can be included in a single folder), with the panels clearly visible in the folder structure instead of a single excel file for all Source Data. e.g. all the Source data files for figure 1 need to be saved in a single folder and this needs to be zipped and then uploaded as "SD figure 1.zip" file.

16) As part of the EMBO Publications transparent editorial process initiative (see our policy here:

https://www.embopress.org/transparent-process#Review_Process), Molecular Systems Biology will publish online a Peer Review File (PRF) to accompany accepted manuscripts. This file will be published in conjunction with your paper and will include the anonymous referee reports, your point-by-point response and all pertinent correspondence relating to the manuscript. Let us know whether you agree with the publication of the PRF and as here, if you want to remove or not any figures from it prior to publication. Please note that the Authors checklist will be published at the end of the PRF.

17) After your paper is published, we may promote it on social media. If you have any handles or hashtags for Bluesky you would like included, please let us know.

18) Please provide a point-by-point letter INCLUDING my comments as well as the reviewer's reports and your detailed responses (as Word file).

I look forward to reading a new revised version of your manuscript as soon as possible.

Yours sincerely,

Poonam Bheda, PhD
Scientific Editor
Molecular Systems Biology

Reviewer #5:

The authors adequately addressed the reviewers criticisms and recommendations. Issues linked to the novelty and impact of the study remain, but the quality of the data analysis is very good and provides interesting conclusions.

Reviewer #6:

As per the editors guidelines, i have only provided comments on whether the authors have responded appropriately to the comments original reviewers.

Reviewers original comments are noted by a bullet point.

Reviewer 1 comment

- The primary weakness of this study is the absence of in vivo data to support the authors' claims. Can the authors observe similar trade-off effects in drug-resistant strains evolved under in vivo conditions (e.g., in mice treated with antifungal drugs)? Additionally, if these resistant strains exhibit fitness trade-offs compared to their parental strain, their infectivity and/or virulence might be reduced in a susceptible host in the absence of drug treatment. Based on the current data, it is unclear to what extent these trade-offs affect the stress resistance and survival of *N. glabrata* within a host. While I acknowledge that such experiments may be practically challenging, the lack of in vivo supporting data undermines the overall significance of this study

The authors responded appropriately to this question

I, in principal agree with the reviewer, although performing such experiments are resource intensive. I am curious as to how the identified mutation correlate to those observed in clinical populations. If these strains are observed clinically, are there known compensatory mechanisms that could potentially circumvent fitness trade offs. I think improved clarity around whether these mutations are clinically relevant would be helpful. The use of *Galleria* goes some way to address the questions around in vivo relevance.

- The authors state that fitness trade-offs are a common feature of drug adaptation (lines 71-72). However, this claim may be overstated, as they did not investigate drug resistance patterns in other fungal pathogens. I am curious whether this phenomenon also occurs in other species, such as *Candida albicans*

The authors responded appropriately to this question

- The authors reported that 98% of drug-resistant strains displayed reduced fitness under at least one stress condition compared to their susceptible parent strains. However, I am curious what percentage of drug-resistant strains demonstrated increased fitness under at least one stress condition

The authors responded appropriately to this question

- The authors suggested that co-targeting the osmotic stress response pathway could prevent the emergence of fluconazole resistance based on the effect of NaCl (lines 841-842). However, they should address the possibility that this strategy may not be universally applicable across all fungal pathogens. For instance, inhibition of the HOG pathway, which regulates osmotic stress response, increases fluconazole resistance in *Cryptococcus neoformans* but enhances fluconazole susceptibility in *Candida albicans*

The authors responded appropriately to this question

I, in principal agree with the reviewer, with their statement. I'm curious how the authors imagine how using NaCl could work therapeutically?

The authors responded appropriately to all additional minor comments

-

Reviewer 2 comment

- The idea that antimicrobial resistance carries a fitness trade-off is well-established in bacterial and fungal systems. The authors fail to convincingly articulate what fundamentally new insights their study provides beyond previous work The authors responded appropriately to this question I, would be curious to know how the results for *N. Glabratus* compare to other fungal pathogens if such studies have previously been performed.

The authors responded appropriately to all subsequent comments

- The term "evolvability" is misapplied in this study. The comparison between fluconazole and fluconazole + NaCl is flawed because cells grown with fluconazole alone undergo more divisions than those in fluconazole + NaCl, which do not reach similar OD levels. Consequently, the conclusions regarding "evolvability" should be toned down

I agree with the reviewer on this claim, and i dont think the experiments are adequately designed to address this. It might be

more appropriate that the combination treatment can select against the emergence of resistance?

- The methods section is overly detailed in some areas (e.g., listing software dependencies) while lacking clarity in key experimental design aspects (e.g., defining "significant" trade-offs)

I agree with the reviewer. Could most of these details be move to supplemental?

-

Reviewer 3 comment

- The dataset used in Fig. 1 appears to include a number of evolved mutants with nearly 0 growth even under YPD. These are not directly discussed in the text, but it is interesting to recognize that they do not seem to be universally poor growing. For instance, all of the ANI evolved mutants grow well in DTT, presumably including these ones. A) At first glance it's unclear why these strains would be included if they barely grow. Do they grow better when anidulafungin is present in the media? B) Can the authors infer any mechanistic connections in the stressful environments based on when these particular strains are able to grow well?

The authors responded appropriately to this and the subsequent question that had a similar line of questioning.

- Figure 3C: I could not determine from the text whether the evolvability assay controls for differences in cell divisions. I might expect cells to be growing more slowly in NaCl (especially in the presence of drug as the authors have pointed out), and reduced cell divisions would be directly explanatory for reduced opportunities to accumulate resistance mutations

I agree with the reviewer on this claim (and as above). A fluctuation assay might be more appropriate to address the claims of evolvability. See additional notes above.

- Figure 2C is mentioned in the text before other components of Figure 2. Figure 4 is mentioned before Figure 3. This complicates following the train of logic

I agree with the reviewer, and dont think the authors have appropriately address this. Convention, is for figure panels to be referenced in the text in order of apperance. The authors should fix this, to improve flow.

- Lines 610-618, supp Fig 2: The authors argue that a synergistic inhibition between NaCl and Fluconazole suggests the mechanisms driving resistance to these stresses overlap. In Figure 1A, increased NaCl sensitivity is observed universally, so one might expect a similar synergistic response with Anidulafungin as well. Following the author's logic, does this mean there is an overlapping response to NaCl stress and Anidulafungin stress? Between Anidulafungin stress and Fluconazole stress? While this may be true, the authors could strengthen their argument by proposing and perhaps investigating a mechanism by which NaCl would cause an overlapping stress with these compounds, in addition to their assertion that chromosome E duplication is associated with reduced fitness in NaCl

Whilst i think the authors have appropriately responded, some element of speculation would be helpful.

It is also worth noting the cidal/static shouldn't effect the "ease" of evolving resistance. Resistance is determine by the number of resistance mutations that are available determining the rate at which they occur. Some highly bactericidal antibiotics (isoniazid and pretomid for TB treatment) have very high frequency of mutations. The experimental setup the authors have used, may also influence the observation of resistance. For example if seeding experiments with $<10^5$ CFUs and exposed to conc $>MIC$, there will be a lack of growth and potentially no pre-existing mutations in the population to be selected. See prior not about fluctuation assays.

-

Reviewer 4 comment

- The work lacks general context in terms of the prior work done on antibiotic resistance and its costs and evolutionary consequences. Even if that is not the focus of the current work, the authors need to be aware that this context is important, particularly for a general interest journal, where a case for broad significance needs to be made by every paper

The authors should try to place this current study in the context of existing literature

Authors have responded appropriately to other comments, although modelling details should be made available even if via a github link.

Manuscript Number: MSB-2025-13139-T

Title: Uncovering actionable trade-offs of antifungal resistance in a yeast pathogen

Dear Dr. Gabaldón,

Thank you for the submission of your revised manuscript to Molecular Systems Biology. Unfortunately we were not able to secure the original reviewers of your manuscript, but we have now received the enclosed reports from two new referees that were asked to assess your point-by-point response to the previous reviewers' concerns and revisions to the manuscript it, overlooking the request for additional in vivo relevance. As you will see the new Reviewer 5 finds the quality of the analysis very good, though both Reviewers 5 and 6 mention that the novelty and impact are an ongoing issue (Reviewer 6 conveyed this via a set of questions that we ask reviewers to answer). Reviewer 6 further comments to individual concerns that for the most part said authors responded appropriately to the reviewer comments, with exceptions that additional analysis of strains that might be observed clinically could improve the in vivo relevance and that additional experiments to appropriately compare strains growing at different rates would address the issues of the term "evolvability" that both Reviewers 2 and 3 had mentioned. Although we are willing to overrule a full experimental investigation into the in vivo relevance and find the novelty sufficient for Molecular Systems Biology, we would ask you to address the remaining concerns by Reviewer 6 in a revision. Please note that regarding the comment that the Methods overly detailed in some places and vague in others, **we would ask you to not move any Methods to the Supplementary Information**, but rather improve the details for any vague descriptions and keep all of the information in the main manuscript as per our policy.

Along with the revision, we would ask you to address the following formatting requirements:

1) Please download the EMBO Press "Author Checklist" and complete all relevant questions. This file should be uploaded with your submission. This file can be downloaded from our website at: <https://www.embopress.org/page/journal/17444292/authorguide>

Response: Done

2) Please include keywords to max. 5.

Response: Done

3) Please include a Data availability section describing how the data, code etc. have been made available. This section needs to be formatted according to the example below:

"The datasets and computer code produced in this study are available in the following databases:

- Chip-Seq data: Gene Expression Omnibus GSE46748
(<https://www.ncbi.nlm.nih.gov/geo/query/acc.cgi?acc=GSE46748>)

- Modeling computer scripts: GitHub
(<https://github.com/SysBioChalmers/GECKO/releases/tag/v1.0>)

- [data type]: [full name of the resource] [accession number/identifier] ([doi or URL or identifiers.org/DATABASE:ACCESSION)]"

Response: done

4) Please include a "Disclosure and competing interests statement". We updated our journal's competing interests policy in January 2022 and request authors to consider both actual and perceived competing interests. Please review the policy <https://www.embopress.org/competing-interests> and update your competing interests if necessary.

Response: done

5) Please remove the heading 'Contributions' from the manuscript and specify author contributions in our submission system. CRediT has replaced the traditional author contributions section because it offers a systematic machine-readable author contributions format that allows for more effective research assessment. You are encouraged to use the free text boxes beneath each contributing author's name to add specific details on the author's contribution. More information is available in our guide to authors: <https://www.embopress.org/page/journal/17574684/authorguide#authorshipguidelines>

Response: done

According to CRediT

Juan Carlos Nunez-Rodríguez (<https://orcid.org/0000-0002-7529-1039>) : Conceptualization, Data curation, Investigation, Methodology, Visualization, Writing – original draft, Writing – review & editing

Miquel Àngel Schikora-Tamarit(<https://orcid.org/0000-0003-2964-9818>) Software, Formal analysis, Methodology, Visualization, Writing – original draft

Toni Gabaldón (<https://orcid.org/0000-0003-0019-1735>) Conceptualization, Funding acquisition, Project administration, Resources, Supervision, Writing – original draft, Writing – review & editing

6) References: Please correct the reference citation in the reference list to be alphabetical (not numerical). Where there are more than 10 authors on a paper, only the first 10 should be listed, followed by "et al.". Please check "Author Guidelines" for more information. <https://www.embopress.org/page/journal/17574684/authorguide#referencesformat>

Response: done

7) Our journal encourages inclusion of *data citations in the reference list* to directly cite datasets that were re-used and obtained from public databases. Data citations in the article text are distinct from normal bibliographical citations and should directly link to the database records from which the data can be accessed. In the main text, data citations are formatted as follows: "Data ref: Smith et al, 2001" or "Data ref: NCBI Sequence Read Archive PRJNA342805, 2017". In the Reference list, data citations must be labeled with "[DATASET]". A data reference must provide the database name, accession number/identifiers and a resolvable link to the landing page from which the data can be accessed at the end of the reference. Further instructions are available at <https://www.embopress.org/page/journal/17574684/authorguide#referencesformat>.

Response: done

8) Please ensure that a statement on whether or not blinding was done is included in the Methods even if no blinding was done. Please also be sure to include the information in the Author Checklist on where it can be found in the manuscript.

Response: done

9) All Materials and Methods need to be described in the main text using our 'Structured Methods' format. According to this format, the Methods section includes a Reagents and Tools Table (listing key reagents, experimental models, software and relevant equipment and including their sources and relevant identifiers) followed by a Methods and Protocols section describing the methods, ideally using a step-by-step protocol format. The aim is to facilitate adoption of the methodologies across labs.

Please download and fill our Reagents and Tools Table template (.docx), which you can find in our author guidelines: <https://www.embopress.org/page/journal/14693178/authorguide#structuredmethods>.

An example of a Method paper with Structured Methods can be found here: <https://www.embopress.org/doi/10.15252/msb.20178071>. "

Response: done

10) Please place individual sections of the manuscript in the following order: Title page - Abstract & Keywords - Introduction - Results - Discussion - Methods - Data Availability - Acknowledgements - Disclosure and Competing Interests Statement - References - Figure Legends - Expanded View Figure Legends.

Response: done

11) For the figures and figure legends, please take care of the following:

- Please remove all figures from main manuscript file and leave only the main figure legends placed after the references.

Response: done

- All figure and panel callouts should be sequential in the manuscript.

Response: done

- Supplemental/Supplementary Figure 1-4 should be renamed to Figure EV1-EV4. Each figure will still need to fit onto one page.

Response: done

Please ensure that the callouts are also updated in the main manuscript. The legends should stay in the manuscript, with the heading Expanded View Figures Legends, and placed after the main figure legends.

Response: done

- Main figures and EV figures should be uploaded as individual, high-resolution files.

Response: done (in vector PDF, resolution 600 DPI)

- Please provide exact p values in the figures or figure legends of Figure 2A, C; S1.

Response: done, exact p values for each comparison are provided together with median values in Dataset_EV1.

- Please note that the box plots need to be defined in terms of minima, maxima, bounds of box and whiskers, and percentile in the legends of figures 2A, C

Response: done

12) Please upload all supplementary tables as one .xsl file per table and rename them to Dataset EV1-5 (filenames and titles within the file). Each dataset will need its legend removed from the manuscript and added to the corresponding file in a separate tab. Please also update their callouts in main manuscript text.

Response: Done

13) Please note that funding information should be given in an "Acknowledgements" section. Please ensure that all funding sources are entered into the manuscript submission system.

Response: Done

14) Synopsis:

- Synopsis image: Please provide a graphic that summarises the main findings of the manuscript on a glance and upload it as a high-resolution jpeg file 550 pixels wide x (300-600) pixels high.

Response: Done

- Synopsis text: Please provide a short standfirst (maximum of 300 characters, including space), limit the bullet points to max. 5 and upload it as a separate .doc file. Please write the bullet points to summarise the key NEW findings. They should be designed to be complementary to the abstract - i.e. not repeat the same text. We encourage inclusion of key acronyms and quantitative information (maximum of 30 words / bullet point). Please use the passive voice.

Response: Done

Response: Done

15) Source Data: Please ensure that a completed Source Data checklist is uploaded as a Related Manuscript File. You will be contacted separately with instructions with the checklist for the Source Data. Source Data should be organized as a single source data file (zipped) per figure for main figures (all EV and/or Appendix figure Source Data can be included in a single folder), with the panels clearly visible in the folder structure instead of a single excel file for all Source Data. e.g. all the Source data files for figure 1 need to be saved in a single folder and this needs to be zipped and then uploaded as "SD figure 1.zip" file.

Response: Done

16) As part of the EMBO Publications transparent editorial process initiative (see our policy here: https://www.embopress.org/transparent-process#Review_Process), Molecular Systems Biology will publish online a Peer Review File (PRF) to accompany accepted manuscripts. This file will be published in conjunction with your paper and will include the anonymous referee reports, your point-by-point response and all pertinent correspondence relating to the manuscript. Let us know whether you agree with the publication of the PRF and as here, if you want to remove or not any figures from it prior to publication. Please note that the Authors checklist will be published at the end of the PRF.

Response: We agree with the publication of the Peer Review File (PRF) carried out by the reviewers of the MSB journal.

17) After your paper is published, we may promote it on social media. If you have any handles or hashtags for Bluesky you would like included, please let us know.

Response: @gabaldonlab.bsky.social

18) Please provide a point-by-point letter INCLUDING my comments as well as the reviewer's reports and your detailed responses (as Word file).

Response: We have addressed each of the editorial and scientific review comments point by point and made modifications to the manuscripts accordingly. To make it easier to read the previous comments and responses, we have indicated the author of each comment and response, and we have marked new responses in red. We have revised the submission format and reviewed the files to ensure they are ready for the proofreading stage.

I look forward to reading a new revised version of your manuscript as soon as possible.

Yours sincerely,

Poonam Bheda, PhD
Scientific Editor
Molecular Systems Biology

Reviewer #5:

The authors adequately addressed the reviewers criticisms and recommendations. Issues linked to the novelty and impact of the study remain, but the quality of the data analysis is very good and provides interesting conclusions.

Response to reviewer 5: We thank reviewer 5 for the appreciation of our work.

Reviewer #6:

As per the editors guidelines, i have only provided comments on whether the authors have responded appropriately to the comments original reviewers.

Reviewers original comments are noted by a bullet point.

#Comment 1:

- **Reviewer 1 comment:** The primary weakness of this study is the absence of in vivo data to support the authors'claims. Can the authors observe similar trade-off effects in drug-resistant strains evolved under in vivo conditions (e.g., in mice treated with

antifungal drugs)? Additionally, if these resistant strains exhibit fitness trade-offs compared to their parental strain, their infectivity and/or virulence might be reduced in a susceptible host in the absence of drug treatment. Based on the current data, it is unclear to what extent these trade-offs affect the stress resistance and survival of *N. glabrata* within a host. While I acknowledge that such experiments may be practically challenging, the lack of in vivo supporting data undermines the overall significance of this study

Author's answer to reviewer 1 comment: We agree on the potential interest of adding in vivo data. However, this work includes a thorough experimental characterisation of the trade-offs in a large collection of resistant strains, as the main focus of our research was to understand the diversity and genetic basis of potential trade-offs related to drug resistance. We consider it impractical to perform in vivo experiments of the same scale, due to practical, economical, and ethical constraints. Experimental vs in-vivo work have different pros and cons and we think both are complementary. We do think our work does provide results that are significant. Nevertheless to explore some of the questions raised by the reviewer we have now tested two serial isolates representing one acquisition of resistance in a patient, and show that the resistant strain shows the NaCl trade-off, underscoring that our findings may reflect processes that can happen in a real clinical situation. Moreover, we have used a *Galleria* model and one of the strains to validate some of our results in-vivo. Furthermore, we believe that the study of trade-off emergence is inherently valuable, as it illuminates the fundamental evolutionary processes at play and informs strategies to combat drug resistance. While certain trade-offs might not directly translate into a competitive disadvantage during in vivo infection (e.g., in humans or mice), their identification is still highly relevant for the development of novel antifungal strategies. Indeed, trade-offs that do not impose a fitness cost during infection are more likely to become established within a population, making them particularly attractive targets for interventions against resistant strains. We have added these clarifications to the discussion section.

Reviewer 6 comment: The authors responded appropriately to this question I, in principal agree with the reviewer, although performing such experiments are resource intensive. I am curious as to how the identified mutation correlate to those observed in clinical populations. If these strains are observed clinically, are there known compensatory mechanisms that could potentially circumvent fitness trade offs. I think improved clarity around whether these mutations are clinically relevant would be helpful. The use of *Galleria* goes some way to address the questions around in vivo relevance.

Response to Reviewer 6: We thank Reviewer 6 for their thoughtful comments and for acknowledging our previous response. The resistance mechanisms observed in our collection, such as mutations in transcription factors like PDR1 or drug targets like ERG11, FKS1, and FKS2, are widely described in clinical settings. Our study also describes chromosome duplications, which are well-known for their relevance in *Candida* species. Furthermore, we describe the potential novel role of ERG3 in resistance, a finding that has recently been supported by other clinical studies (<https://doi.org/10.1128/mbio.01419-25>).

However, the specific mutations within these genes are less described, given the wide diversity and scale of mutations observed. This limited overlap is also evident in large clinical

surveys (e.g., SENTRY), which show that the number of possible mutations (i.e., the precise residue affected and the type of mutation) is vast and highly diverse, and thus currently only partially captured by either our study or clinical datasets. Importantly, our findings demonstrate that regardless of the specific mutations acquired, trade-offs are frequently observed. Therefore, if such effects arise in this highly diverse experimental collection, it is reasonable to expect that they will also occur in other clinical isolates. As part of our revisions, we included the analysis of a clinical serial isolate carrying a PDR1 mutation, where we validated this concept by detecting a trade-off in response to NaCl.

In accordance with the reviewer's recommendation, we have expanded the introduction to include a more thorough description of the collection and its clinical relevance, providing better context.

With respect to compensatory mechanisms, while these are well documented in bacteria, they remain less explored in *Candida*, particularly regarding responses to stressors. Nonetheless, recent work has shown a link between amphotericin B resistance, oxidative stress, and fitness in *C. auris*, where mutations in CDC25 were reported to alleviate the fitness trade-offs caused by ERG6 mutations (<https://doi.org/10.1038/s41564-024-01854-z>). We now discuss this further in the manuscript.

Finally, we are grateful for the Reviewer's positive assessment of the inclusion of the *Galleria* model experiment, which we believe further strengthens the relevance of our study.

—

#Comment 2:

- **Reviewer 1 comment:** The authors state that fitness trade-offs are a common feature of drug adaptation (lines 71-72). However, this claim may be overstated, as they did not investigate drug resistance patterns in other fungal pathogens. I am curious whether this phenomenon also occurs in other species, such as *Candida albicans*

Response of authors to reviewer 1: We acknowledge that it would be valuable to explore this phenomenon in other species. Nevertheless, the core of our study lies in providing an in-depth understanding of what occurs within the considerable diversity of *Nakaseomyces glabratus*. We've made sure to emphasize this species-specific focus throughout the text.

Reviewer 6 comment: The authors responded appropriately to this question

Response to reviewer 6: Thanks.

—

#Comment 3:

- **Reviewer 1 comment:** The authors reported that 98% of drug-resistant strains displayed reduced fitness under at least one stress condition compared to their

susceptible parent strains. However, I am curious what percentage of drug-resistant strains demonstrated increased fitness under at least one stress condition

Response of authors to reviewer 1: We've now included this information in the updated Supplemental Table 4. This table shows the list of strains that exhibited statistically significant increases in fitness, using the same thresholds applied for the negative trade-offs.

Reviewer 6 comment: The authors responded appropriately to this question

Response to reviewer 6: Thanks.

—

#Comment 4:

- **Reviewer 1 comment:** The authors suggested that co-targeting the osmotic stress response pathway could prevent the emergence of fluconazole resistance based on the effect of NaCl (lines 841-842). However, they should address the possibility that this strategy may not be universally applicable across all fungal pathogens. For instance, inhibition of the HOG pathway, which regulates osmotic stress response, increases fluconazole resistance in *Cryptococcus neoformans* but enhances fluconazole susceptibility in *Candida albicans*

Response of authors to reviewer 1: We agree that these results are specific for *Nakaseomyces glabratus*, and we have emphasized this fact throughout the text.

Reviewer 6 comment: The authors responded appropriately to this question. I, in principal agree with the reviewer, with their statement. I'm curious how the authors imagine how using NaCl could work therapeutically?

Response to reviewer 6: We thank the reviewer for this insightful question. Our intention was not to propose NaCl as a direct therapeutic agent, as we agree this would not be clinically feasible for invasive diseases. Instead, we used NaCl sensitivity, which was the most commonly identified trade-off, to derive a proof-of-concept that trade-offs arising with drug resistance create specific vulnerabilities that can be strategically exploited. NaCl sensitivity signals a vulnerability in the osmotic/ionic stress response. Beyond NaCl, one could envision drugs that exploit this specific vulnerability, for instance, by targeting ion transporters, inducing a selective ionic imbalance with ionophores, or targeting membrane stress response mechanisms even if these are not yet a clinical reality. In line with these ideas a recent paper from Leah Cowen lab (<https://doi.org/10.1038/s41467-022-31308-1>) describes imidazopyrazoindole as a compound that targets fungal membrane homeostasis to induce membrane-stress, showing high efficiency in combination with fluconazole .

In response to the reviewer's suggestion, we have improved the discussion with some comments on possible applications, as well as on the mechanistic hypothesis of this interaction that has been suggested in other comments.

#Comment 5:

Reviewer 6 comment: The authors responded appropriately to all additional minor comments

Response to reviewer 6: Thank you.

#Comment 6:

- **Reviewer 2 comment:** The idea that antimicrobial resistance carries a fitness trade-off is well-established in bacterial and fungal systems. The authors fail to convincingly articulate what fundamentally new insights their study provides beyond previous work

Response of authors to reviewer 2: We agree with the reviewer that fitness trade-offs are a well-established concept across various biological systems. However, we believe there's a significant knowledge gap regarding their generality and prevalence specifically within resistant to drugs in fungal pathogens.

Furthermore, our research offers several fundamentally new insights. We conducted experiments on a large collection under different conditions and replicates, generating a large and highly relevant dataset on trade-offs in *N. glabratus*. This extensive characterization provides a robust foundation for understanding these phenomena. We applied innovative machine learning models to our data that allowed us to identify potential underlying causes and mechanisms driving these trade-offs, moving beyond mere observation to mechanistic insights in this highly complex type of interactions. Crucially, we also demonstrate the concept that studying these trade-offs can directly lead to the proposal of novel intervention strategies against drug-resistant strains. In our opinion, we consider that we have significantly contributed to the knowledge of this field and laid certain foundations so that it can be expanded in the future.

Reviewer 6 comment: The authors responded appropriately to this question I, would be curious to know how the results for *N. Glabratus* compare to other fungal pathogens if such studies have previously been performed.

Response to reviewer 6: We thank Reviewer 6 for the positive assessment of our previous response. It is true that the study of stress-related trade-offs has not been widely explored in fungal pathogens; however, recent work increasingly highlights the relevance of this process. In particular, novel studies in *C. auris* have reported findings consistent with our results in *N. glabratus*, including reduced tolerance to H₂ O₂ associated with fluconazole resistance, increased sensitivity to CFW in echinocandin-resistant strains, and decreased fitness in response to stressors similar to those tested in our study in amphotericin B-resistant strains. Also we cited drug resistance trade-offs described in *C. parapsilosis*. We now cite and discuss these studies in the revised manuscript.

#Comment 7:

Reviewer 6 comment: The authors responded appropriately to all subsequent comments

Response to reviewer 6: Thank you.

—

#Comment 8:

- **Reviewer 2 comment:** The term "evolvability" is misapplied in this study. The comparison between fluconazole and fluconazole + NaCl is flawed because cells grown with fluconazole alone undergo more divisions than those in fluconazole + NaCl, which do not reach similar OD levels. Consequently, the conclusions regarding "evolvability" should be toned down

Response of authors to reviewer 2: We are open to rephrasing some of our claims to better reflect the nuances of our data. While we acknowledge that cells grown with fluconazole alone undergo more divisions, we still believe our data demonstrates a mechanistic link between fluconazole and NaCl that underlies the lack of resistance acquisition in the presence of NaCl. The progressive increase in fluconazole resistance in the fluconazole-only condition highlights that resistance acquisition under these conditions is highly frequent, allowing strains to grow in concentrations up to 256 ug/ml within just four passages. A lower number of divisions in the fluconazole + NaCl condition might lead to a softer resistance acquisition curve. However, this difference in division rate isn't significant enough to fully explain the complete absence of resistance acquisition in those strains.

Reviewer 6 comment: I agree with the reviewer on this claim, and i dont think the experiments are adequately designed to address this. It might be more appropriate that the combination treatment can select against the emergence of resistance?

Response to reviewer 6: In line with Reviewer 6's comment, we agree that there is solid evidence supporting the idea that the combination of NaCl and fluconazole selects against the emergence of resistance. To better reflect this concept, we have revised the terminology and now use the term "adaptability," defined as the ability to adapt to a specific condition. In addition, we have further explored the possible influence of a reduced number of divisions, which we discuss in a later comment.

Furthermore, we have performed new experiments, now reflected in Figure EV3C and 3D, which follow the principle suggested in your comment. Specifically, we plated 10^7 cells directly onto single and combination treatments of NaCl and fluconazole. We found that the probability of directly isolating mutants capable of overcoming the combined stress is less than 10^7 .

In contrast, we were able to find a small proportion of mutants that could withstand the NaCl stress after pre-selecting for fluconazole-adapted colonies. These new data strongly support our previous findings, reinforcing the idea that identifying shared trade-offs is a promising strategy for developing treatments to reduce the acquisition of drug resistance, and specifically the connection between NaCl and fluconazole.

These modifications have been incorporated into the revised manuscript.

—

#Comment 9:

- **Reviewer 2 comment:** The methods section is overly detailed in some areas (e.g., listing software dependencies) while lacking clarity in key experimental design aspects (e.g., defining "significant" trade-offs)

Reviewer 6 comment: I agree with the reviewer. Could most of these details be move to supplemental?

Response: The editor requested not to do that. Details are necessary to ensure reproducibility. We revised the methods to ensure sufficient clarity was provided.

—

#Comment 10:

- **Reviewer 3 comment:** The dataset used in Fig. 1 appears to include a number of evolved mutants with nearly 0 growth even under YPD. These are not directly discussed in the text, but it is interesting to recognize that they do not seem to be universally poor growing. For instance, all of the ANI evolved mutants grow well in DTT, presumably including these ones. A) At first glance it's unclear why these strains would be included if they barely grow. Do they grow better when anidulafungin is present in the media? B) Can the authors infer any mechanistic connections in the stressful environments based on when these particular strains are able to grow well?

Response of authors to reviewer 3: We acknowledge that this point may initially appear to be confusing. However, the figures illustrate relative growth (AUC_R or fAUC_R), as detailed comprehensively in both the Methods section and Figure 1D. It is important to remark that a relative growth value of 1, indicating similar relative growth (vs rich media) under stress compared to the parental strain (fAUC_R), does not signify a recovery of absolute growth capacity. If the growth in rich medium is inherently low, and this low growth is maintained under a specific stress condition, the resulting fAUC_R will be close to 1. This approach ensures that our analysis accurately reflects specific stress-induced trade-offs rather than overall growth deficiencies.

Reviewer 6 comment: The authors responded appropriately to this and the subsequent question that had a similar line of questioning.

Response to Reviewer 6: Thank you.

—

#Comment 11:

- **Reviewer 3 comment:** Figure 3C: I could not determine from the text whether the evolvability assay controls for differences in cell divisions. I might expect cells to be

growing more slowly in NaCl (especially in the presence of drug as the authors have pointed out), and reduced cell divisions would be directly explanatory for reduced opportunities to accumulate resistance mutations

Response of authors to reviewer 3: While we agree there might be a difference in the number of cell divisions between the conditions, as we've discussed previously, we don't believe this difference is significant enough to fully explain the complete absence of resistance acquisition when NaCl is present. In line with that, under drug-only conditions, where initial cell divisions are also impacted, the wild-type strains still efficiently increase their resistance. This suggests that while division rate plays a role, the presence of NaCl introduces a more fundamental barrier to resistance evolution, rather than simply reducing the opportunities for mutations.

Reviewer 6 comment: I agree with the reviewer on this claim (and as above). A fluctuation assay might be more appropriate to address the claims of evolvability. See additional notes above.

Response to Reviewer 6: We appreciate Reviewer 6's comment and understand the concerns raised about this experiment. First, we have revised our terminology and now use the term "adaptability," which we believe better reflects the concept supported by our experimental data. Second, we have addressed in more depth the important point raised by Reviewer 3 regarding differences in cell divisions.

To evaluate this, we revisited the growth data (OD600), which are fully available in Dataset EV5. As expected, growth in YPD is higher than in YPD + NaCl (1.25 M), indicating that strains experience some inhibition at this concentration, though they are still able to substantially grow. To assess whether this inhibition could drastically reduce the number of divisions and thereby impact adaptation probabilities, we estimated the approximate number of generations during the experiment. We applied a Gompertz-based calculation:

$$N = N_0 \times 2^g \rightarrow g = \log_2(N/N_0),$$

where g represents the number of generations, N is the final cell number (approximated from final OD), and N₀ corresponds to the starting inoculum (dilution 1/200 from the OD of the previous MGC well). As an example we show here the calcs for number of generations for the CBS138 biological replicate 2 (CBS138-R2) under fluconazole alone and NaCl + fluconazole (the replicate shown in Figure 3C).

Day	Condition	MGC	Final OD (MGC)	Initial OD to next passage (dilution 1/200)	g=Log2(Final OD/Initial OD)
Day 1	flz	8	1.14	0.0057	
Day 2	flz	16	0.862	0.0043	7.241
Day 3	flz	64	0.976	0.0049	7.823
Day 4	flz	128	0.817	0.0041	7.387
Day 1	NaCl+flz	1	0.649	0.0032	
Day 2	NaCl+flz	4	0.647	0.0032	7.639
Day 3	NaCl+flz	2	0.464	0.0023	7.164
Day 4	NaCl+flz	4	0.653	0.0033	8.137

The final number of generations was very similar: 7-8 per passage (24 h). The apparent discrepancy between lower final growth (OD) in NaCl and the similar number of generations arises because inocula differ slightly across conditions: higher in fluconazole-only passages

(which previous passages were growing more) and lower in NaCl + fluconazole (which previous passages were growing less). Thus, while final cell number (OD) differs, the number of generations remains comparable.

Although minor differences in total cell numbers may affect the absolute probability of resistance emergence, they do not account for the striking observation that under fluconazole alone, strains consistently and exponentially increase resistance (up to 256 µg/ml), while under NaCl + fluconazole they completely fail to increase resistance after four days of exposure.

Third, as previously mentioned, and in line with the reviewer's suggestion for a fluctuation assay, we have performed new experiments (now included in Figure EV3C and 3D). We used a high-density plating (10^7) approach and replica plating from fluconazole-selected strains to demonstrate that the double bottleneck (the requirement to be fluconazole-resistant and lack the NaCl trade-off) significantly reduces the probability of selecting fluconazole-resistant strains.

Together with the demonstrated synergy between NaCl and fluconazole, and the fact that NaCl represents the major trade-off identified in fluconazole-resistant strains, this strongly supports the original conclusion that the reduced adaptability to fluconazole is specifically mediated by NaCl, rather than merely by reduced opportunities for cell divisions.

Accordingly, we have revised the manuscript to ensure that the claims are closely aligned with the experimental data, replaced the term "evolvability" with "adaptability," changed the conceptual image in Figure 3A, and added the results of the new experiments in Figures EV3C and 3D.

#Comment 12:

- **Reviewer 3 comment:** Figure 2C is mentioned in the text before other components of Figure 2. Figure 4 is mentioned before Figure 3. This complicates following the train of logic

Response of authors to reviewer 3: Response: For Figure 2, the current ordering was chosen to best align with the visual structure of the figure's components. Regarding Figure 4B, as we've discussed previously, this panel presents data from a competition assay that encompasses various conditions. These conditions culminate with those explained in the section pertaining to Figure 4. Presenting this data earlier would lack the necessary context established in the preceding sections. Nevertheless, we are open to revising the figure order or placement if the reviewers or editors believe it would enhance the logical flow of the manuscript.

Reviewer 6 comment: I agree with the reviewer, and don't think the authors have appropriately address this. Convention, is for figure panels to be referenced in the text in order of appearance. The authors should fix this, to improve flow.

Response to Reviewer 6: As suggested by the reviewers, we ensure that figure references in the text follow a consistent and logical order. We have moved this panel from Figure 4 to the Figure EV2

—

#Comment 13:

- **Reviewer 3 comment:** Lines 610-618, supp Fig 2: The authors argue that a synergistic inhibition between NaCl and Fluconazole suggests the mechanisms driving resistance to these stresses overlap. In Figure 1A, increased NaCl sensitivity is observed universally, so one might expect a similar synergistic response with Anidulafungin as well. Following the author's logic, does this mean there is an overlapping response to NaCl stress and Anidulafungin stress? Between Anidulafungin stress and Fluconazole stress? While this may be true, the authors could strengthen their argument by proposing and perhaps investigating a mechanism by which NaCl would cause an overlapping stress with these compounds, in addition to their assertion that chromosome E duplication is associated with reduced fitness in NaCl

Response of authors to reviewer 3: Our preliminary data investigating the interactions between anidulafungin and NaCl were not conclusive. This was primarily because the rate of anidulafungin resistance acquisition was very low due to its fungicidal, rather than fungistatic, effect.

On the other hand, as the reviewer rightly pointed out, we did find a significant correlation between Chromosome E duplications and reduced fitness in NaCl. However, despite analyzing dozens of strains with observed defects, we haven't been able to identify enough conclusive features to propose a specific mechanism explaining this phenomenon. Our results point towards complex epistatic relationships that are highly background-dependent. This inherent complexity currently prevents us from proposing a clear, singular mechanism for these observed effects.

Reviewer 6 comment: Whilst i think the authors have appropriately responded, some element of speculation would be helpful.

It is also worth noting the cidal/static shouldn't effect the "ease" of evolving resistance. Resistance is determine by the number of resistance mutations that are available determining the rate at which they occur. Some highly bactericidal antibiotics (isoniazid and pretomnid for TB treatment) have very high frequency of mutations. The experimental setup the authors have used, may also influence the observation of resistance. For example if seeding experiments with $<10^5$ CFUs and exposed to conc $>MIC$, there will be a lack of growth and potentially no pre-existing mutations in the population to be selected. See prior not about fluctuation assays.

Response to Reviewer 6: We thank Reviewer 6 for their thoughtful feedback and agree that the fungistatic or fungicidal nature of a drug does not directly determine the mutation rate underlying resistance emergence. Nonetheless, in *Candida* spp. it has been reported that tolerance to fungistatic drugs is generally higher than to fungicidal drugs (<https://doi.org/10.1038/s41467-018-04926-x>), and that tolerance can increase the likelihood of resistance acquisition (<https://doi.org/10.1099/acmi.cc2021.po0031>). In any case, these considerations were not included in the manuscript.

We have already discussed our considerations regarding the adaptability test above and we performed the experiment suggested by the reviewer seeding 10^7 CFU in 8 fold-increase MIC, the results reinforce our previous data and hypothesis. Following the reviewer's suggestion, we have now added discussion on potential mechanisms underlying the overlap between drug responses and NaCl stress. While the precise molecular connections remain to be determined, our findings, together with accumulating evidence in the field, point toward complex and interconnected mechanisms of stress response and drug resistance.

—

#Comment 14:

- **Reviewer 4 comment:** The work lacks general context in terms of the prior work done on antibiotic resistance and its costs and evolutionary consequences. Even if that is not the focus of the current work, the authors need to be aware that this context is important, particularly for a general interest journal, where a case for broad significance needs to be made by every paper

Response of authors to reviewer 4: We appreciate the comment on the broader context of our work. We initially aimed for a concise format to align with the editorial style of the journal. However, we agree to strengthen the context also in prokaryotes if deemed necessary.

Reviewer 6 comment: The authors should try to place this current study in the context of existing literature

Response to Reviewer 6: We have now expanded the manuscript introduction to provide broader context, placing our study within the framework of previous work on bacteria and other organisms.

—

#Comment 15:

Reviewer 6 comment: Authors have responded appropriately to other comments, although modelling details should be made available even if via a github link.

Response to Reviewer 6: We thank Reviewer 6 for their constructive feedback throughout the review process. All the code is now available at https://github.com/Gabaldonlab/Cglabrata_tradeoffs, and we have provided detailed descriptions of the modeling procedures in the Methods section. In addition, we have included the source data for Figure 5, containing all parameters, performance metrics, and selected features, to ensure full interpretability and reproducibility.

11th Dec 2025

Manuscript Number: MSB-2025-13139R

Title: Uncovering actionable trade-offs of antifungal resistance in a yeast pathogen

Dear Dr. Gabaldón,

Thank you for the submission of your revised manuscript to Molecular Systems Biology. We have now received the enclosed report from Reviewer 6 that was asked to re-assess it. Based on the overall reviews, I am pleased to inform you that we will be able to accept your manuscript pending the following final amendments:

- The reference citations are still not correctly formatted. Please ensure that when there are more than 10 authors on a paper, only the first 10 should be listed, followed by "et al.". We note that "et al." was not included in the author lists.
- Please note that full funding information should be given in both the manuscript as well as in our online submission system. Currently the following are still missing information in our submission system eJP: the Spanish Ministry of Science and Innovation for grants CPP2021-008552 and PDC2022-133266-I00, cofounded by ERDF "A way of making Europe"; from the Catalan Research Agency (AGAUR) SGR01551; from the European Union's Horizon 2020 research and innovation programme (ERC-2016-724173); from the Gordon and Betty Moore Foundation (Grant GBMF9742); the Instituto de Salud Carlos III (IMPACT Grant IMP/00019 and CIBERINFEC CB21/13/00061- ISCIII-SGEFI/ERDF); a Predoctoral Fellowship from the Spanish Ministry of Science and Innovation (grant number PRE2019-088193); a Predoctoral Fellowship from the 'La Caixa' Foundation (grant number LCF/BQ/DR19/11740023).
- Please ensure that the synopsis image is uploaded in the correct format - not a PDF, but rather as a high-resolution jpeg file 550 pixels wide x (300-600) pixels high. In addition, we would suggest that you simplify the synopsis image, as the text will be difficult to read in the small size of the figure. Moreover, we would advise a graphic that summarises the main findings of the manuscript on a glance.

Yours sincerely,

Poonam Bheda, PhD
Scientific Editor
Molecular Systems Biology

Reviewer #6:

The authors have addressed my prior comments.

Whilst there is an ever increasing body of work describing the tradeoffs of drug resistance, the current study nicely translates these concepts into pathogenic fungi and the integration of clinically relevant strains.

- The reference citations are still not correctly formatted. Please ensure that when there are more than 10 authors on a paper, only the first 10 should be listed, followed by "et al.". We note that "et al." was not included in the author lists.

Response: We have modified the references according to EMBO press criteria and ensured that where there are more than 10 authors on a paper, only the first 10 are listed, followed by 'et al.'. We have also updated a reference that was in preprint and has now been published (Carolus et al, 2025).

- Please note that full funding information should be given in both the manuscript as well as in our online submission system. Currently the following are still missing information in our submission system eJP:

Response: We have included all funding information in the system.

- Please ensure that the synopsis image is uploaded in the correct format - not a PDF, but rather as a high-resolution jpeg file 550 pixels wide x (300-600) pixels high. In addition, we would suggest that you simplify the synopsis image, as the text will be difficult to read in the small size of the figure. Moreover, we would advise a graphic that summarises the main findings of the manuscript on a glance.

Response: We have simplified the image and increased the text size to make it legible. We have added the image in 500x370 px in JPG format, although due to these pixels the quality is not high definition. We can send the image in vector format or with the same size ratio but with a higher dpi to increase its quality in JPG format.

15th Dec 2025

Manuscript number: MSB-2025-13139RR

Title: Uncovering actionable trade-offs of antifungal resistance in a yeast pathogen

Dear Dr. Gabaldón,

Thank you again for sending us your revised manuscript. We are now satisfied with the modifications made and I am pleased to inform you that your paper has been accepted for publication.

You may qualify for financial assistance for your publication charges - either via a Springer Nature fully open access agreement or an EMBO initiative. Check your eligibility: <https://link.springer.com/journal/44320/how-to-publish-with-us>

Yours sincerely,

Sincerely,

Poonam Bheda, PhD
Scientific Editor
Molecular Systems Biology

>>> Please note that it is Molecular Systems Biology policy for the transcript of the editorial process (containing referee reports and your response letter) to be published as an online supplement to each paper. If you do NOT want this, you will need to inform the Editorial Office via email immediately. More information is available here: <https://link.springer.com/partners/embo-press/editorial-policies#Peer%20review>